# GraphGini: Fostering Individual and Group Fairness in Graph Neural Networks

**Anuj Kumar Sirohi**[*]  *aiz218324@iitd.ac.in*
*Indian Institute of Technology Delhi*
**Anjali Gupta**[*]  *anjali@cse.iitd.ac.in*
*Indian Institute of Technology Delhi*
**Sandeep Kumar**  *ksandeep@iitd.ac.in*
*Indian Institute of Technology Delhi*
**Amitabha Bagchi**  *bagchi@cse.iitd.ac.in*
*Indian Institute of Technology Delhi*
**Sayan Ranu**  *sayanranu@cse.iitd.ac.in*
*Indian Institute of Technology Delhi*

**Reviewed on OpenReview:** *https://openreview.net/forum?id=IEVGBI9MiL*

## Abstract

Graph Neural Networks (GNNs) have demonstrated impressive performance across various tasks, leading to their increased adoption in high-stakes decision-making systems. However, concerns have arisen about GNNs potentially generating unfair decisions for underprivileged groups or individuals when lacking fairness constraints. This work addresses this issue by introducing GRAPHGINI, a novel approach that incorporates the *Gini coefficient* to enhance both individual and group fairness within the GNN framework. We rigorously establish that the Gini coefficient offers greater robustness and promotes equal opportunity among GNN outcomes, advantages not afforded by the prevailing Lipschitz constant methodology. Additionally, we employ the *Nash social welfare program* to ensure our solution yields a Pareto optimal distribution of group fairness. Extensive experimentation on real-world datasets demonstrates GRAPHGINI's efficacy in significantly improving individual fairness compared to state-of-the-art methods while maintaining utility and group fairness.

## 1 Introduction and Related Works

Graph Neural Networks (GNNs) are increasingly being adopted for various high-stakes applications, including credit scoring for loan issuance (Shumovskaia et al., 2020), medical diagnosis (Ahmedt-Aristizabal et al., 2021), molecular property prediction (Wieder et al., 2020; Sirohi et al., 2025; Nishad et al., 2020), and recommendation engines and (Fan et al., 2019; Gong et al., 2020; Agrawal et al., 2024; Gupta et al., 2025; V et al., 2025; Chakraborty et al., 2023; Manchanda et al., 2020). However, recent studies have shown that GNNs may explicitly or implicitly inherit existing societal biases in the training data and, in turn, generate decisions that are socially unfair (Dong et al., 2023; Dai et al., 2022). It is, therefore, important to guard GNNs from being influenced by sensitive features such as gender, race, religion, etc.

There are two primary approaches to address the issue of algorithmic bias: *individual fairness* and *group fairness* (Mehrabi et al., 2021; Liu et al., 2022). Individual fairness involves masking the sensitive features and ensuring that individuals who are alike with respect to non-sensitive features receive similar treatment or outcomes from a system. The group fairness approach, on the other hand, looks within groups defined by sensitive features, e.g., female job applicants or male job applicants, and works to ensure that each such group has similar individual fairness characteristics to all other groups. For example, the variability in treatment between two highly qualified female candidates should be no more than the variability in treatment between

---

[*]These authors contributed equally to this work.

two highly qualified male candidates. In general, a fair system should ensure *both* individual fairness and group fairness since they ensure parity in two complementary dimensions.

## 1.1 Existing Works

**Individual fairness:** For individual fairness, the key optimization parameter sought to be minimized is the *Lipschitz constant*, i.e. the worst-case ratio of the distance between the GNN-generated embeddings of pairs of points and their distance based on some domain-specific criteria (e.g., number of common friends, similarity on initial attributes, etc.). The advantage of this metric, as observed in several works (c.f., e.g., (Kang et al., 2020; Lahoti et al., 2020; Dwork et al., 2012; Song et al., 2022)), is that it is differentiable. Intuitively, a small Lipschitz constant implies that the distance in the embedding space is similar to the distance in the input space. Since the GNN prediction is a function of the embeddings, this implies that similar individuals are likely to get similar predictions (Kang et al., 2020).

**Group fairness:** Group fairness in GNNs has the objective of achieving accurate prediction while also being independent of *protected* attributes that define groups. Rahman et al. (2019) propose the idea of "equality of representation", which expands upon the concept of statistical parity for the node2vec model. Bose & Hamilton (2019), and Dai & Wang (2021) propose adversarial approaches to eliminate the influence of sensitive attributes. He et al. (2023) enables group fairness with respect to multiple non-binary sensitive attributes. Among other works, the strategies include balance-aware sampling (Lin et al., 2023), attention to mitigate group bias (Kose & Shen, 2023) and bias-dampening normalization (Kose & Shen, 2022; Lin et al., 2024; Yang et al., 2024).

***Both* group and individual fairness:** GUIDE Song et al. (2022) is the first and only state-of-the-art work to integrate group and individual fairness principles within a GNN framework. Using Lipschitz constant minimization to enforce individual fairness, GUIDE sought to achieve group fairness by trying to equalize the individual fairness achieved across groups. More recent works, such as Wang et al. (2024) and Yan et al. (2024), further explore this intersection by proposing alternative fairness formulations and optimization strategies to simultaneously improve both fairness notions.

## 1.2 Contributions

The Lipschitz constant, because it is differentiable, has been the predominant metric for optimizing both individual and group fairness. However, this mathematical convenience comes at a cost. Specifically, the Lipschitz constant, being a *max* operator, fails to capture the distribution of outcomes, making it overly restrictive and sensitive to outliers. To address its restricted expressivity, we introduce GRAPHGINI, incorporating three key innovations:

- **Incorporating Gini and establishing its superior encapsulation of fairness:** To capture fairness on the entire spectrum of outcomes, we propose the *Gini coefficient*, a well-established social welfare metric. We show that the Gini coefficient offers a more robust and holistic fairness measure, naturally leading to *equal opportunity* Hardt et al. (2016).

- **Differential approximation of the Gini coefficient:** Fair ML models, including GNNs, typically enforce fairness constraints as regularizers in the loss function. Unlike the Lipschitz constant, the Gini coefficient is non-differentiable, preventing direct optimization. We address this by proving an upper bound that is differentiable, allowing its seamless incorporation as a loss regularizer.

- **Pareto Optimality:** We establish a *Pareto optimal* distribution of group fairness through the use of the *Nash Social Welfare Program* (NSWP) (Charkhgard et al., 2022), which is an optimization technique that provides provably Pareto optimal solutions for multi-objective optimization problems.

With the incorporation of the proposed innovations, GRAPHGINI, outperforms 11 state-of-the-art fair GNN algorithms in both individual and group fairness, with minimal impact on utility. We demonstrate this using an empirical framework that evaluates 3 GNN backbones across three real-world datasets.

## 2 Background and Problem Formulation

We use bold uppercase letters (e.g., $\mathbf{A}$) to denote matrices and $\mathbf{A}[\mathbf{i},:]$, $\mathbf{A}[:,\mathbf{j}]$ and $\mathbf{A}[\mathbf{i},\mathbf{j}]$ represent the $i$-th row, $j$-th column, and $(i,j)$-th entry of a matrix $\mathbf{A}$ respectively. For notational brevity, we use $\mathbf{z_i}$ to represent the row vector of a matrix, i.e., $\mathbf{z_i} = \mathbf{Z}[\mathbf{i},:]$). As convention, we use lowercase (e.g., $n$) and bold lowercase (e.g., $\mathbf{z}$) variable names to denote scalars and vectors, respectively. We use $\mathrm{Tr}(\mathbf{A})$ to denote the trace of matrix $\mathbf{A}$. The $\ell_1$ and $\ell_2$-norm of a vector $\mathbf{z} \in \mathbb{R}^d$ are defined as $||\mathbf{z}||_\mathbf{1} = \sum_{\mathbf{i=1}}^{\mathbf{d}} |\mathbf{z_i}|$ and $||\mathbf{z}||_2 = \sqrt{\sum_{i=1}^{d} z_i^2}$ respectively. All notations are summarized in Table J in the Appendix.

**Definition 1** (Graph). *A graph $G = (\mathcal{V}, \mathcal{E}, \mathbf{X})$ has (i) $n$ nodes ($|\mathcal{V}| = n$) (ii) a set of edges $\mathcal{E} \in \mathcal{V} \times \mathcal{V}$ and (iii) a feature matrix $\mathbf{X} \in \mathbb{R}^{n \times d}$ characterizing each node.*

**Definition 2** (Graph Neural Network (GNN) and Node Embeddings). *A graph neural network consumes a graph $G = (\mathcal{V}, \mathcal{E}, \mathbf{X})$ as input, and embeds every node into a $c$-dimensional feature space. We denote $\mathbf{Z} \in \mathbb{R}^{n \times c}$ to be the set of embeddings, where $\mathbf{z}_i$ denotes the embedding of node $v_i \in \mathcal{V}$.*

**Definition 3** (Sensitive attributes). *Sensitive attributes are those attributes that should not be used by the GNN to influence the prediction (e.g. gender). As a function of these attributes, $\mathcal{V}$ may be partitioned into disjoint groups.*

For instance, gender may be classified as a sensitive attribution, which partitions the node set into $\mathcal{V}_{male}$ and $\mathcal{V}_{female}$ such that $\mathcal{V} = \mathcal{V}_{male} \cup \mathcal{V}_{female}$.

**Definition 4** (User similarity matrix $\mathbf{S}$ and Laplacian $\mathbf{L}$). *Based on domain knowledge/application, the similarity matrix $\mathbf{S}$ denotes the similarity across any pair of nodes. The Laplacian $\mathbf{L}$ of $\mathbf{S}$ is defined as $\mathbf{L} = \mathbf{D} - \mathbf{S}$ where, $\mathbf{D}$ is a diagonal matrix, with $\mathbf{D}[\mathbf{i},\mathbf{i}] = \sum_{j=1,j\neq i}^{n} \mathbf{S}[\mathbf{i},\mathbf{j}]$.*

Without loss of generality, we assume the similarity value for a pair of nodes lies in the range $[0,1]$. The *distance* between nodes is defined to be the inverse of similarity, i.e., for two nodes $u_i, u_j \in \mathcal{V}$, $d_{i,j} = \frac{1}{\mathbf{S}[\mathbf{i},\mathbf{j}]+\delta}$, where $\delta$ is a small positive constant to avoid division by zero. Furthermore, we assume that the distance/similarity is symmetric and is computed after masking the sensitive attributes.

### 2.1 Individual fairness

Individual fairness demands that any two similar individuals should receive similar algorithmic outcomes (Kang et al., 2020). In our setting, this implies that if two nodes ($v_i$ and $v_j$) are similar (i.e., $\mathbf{S}[v_i, v_j]$ is high), their embeddings ($\mathbf{z}_i$ and $\mathbf{z}_j$) should be similar as well.

As discussed in § 1, the Lipschitz constant has been the predominant choice for enforcing individual fairness. In this section, we concretely establish its limitations and show how the Gini coefficient offers a more expressive and robust mechanism for modeling individual fairness.

#### 2.1.1 Limitations of the Lipschitz Constant

**Definition 5.** *The Lipschitz constant of a GNN with node embeddings $\mathbf{Z} \in \mathbb{R}^{n \times c}$ is defined as the smallest constant $L$ such that for all pairs of nodes $v_i, v_j \in \mathcal{V}$:*

$$\|\mathbf{z}_i - \mathbf{z}_j\|_1 \leq L \cdot d_{i,j} \implies L \geq \max_{\forall v_i, v_j \in \mathcal{V}} \left\{ \frac{\|\mathbf{z}_i - \mathbf{z}_j\|_1}{d_{i,j}} \right\} \tag{1}$$

Since the Lipschitz constant is defined using a max operator, it only captures the worst-case discrepancy between input distances and GNN embeddings, ignoring the full distribution of variations across all node pairs. This results in an incomplete and potentially misleading quantification of inequality. We illustrate this

with a concrete example over a population of 5 nodes, whose distance matrix is as follows.

|   | A | B | C | D | E |
|---|---|---|---|---|---|
| A | 1 | 2 | 2 | 2 | 2 |
| B |   | 1 | 2 | 2 | 2 |
| C |   |   | 1 | 2 | 2 |
| D |   |   |   | 1 | 2 |
| E |   |   |   |   | 1 |

We only show the upper triangle due to symmetry of the similarity function. The distance between a pair of nodes $i, j$ is defined as its inverse, i.e., $d_{ij} = \frac{1}{\mathbf{S}[i,j]}$. Thus, in our case, all pairs of distinct nodes have a distance of 2.

Corresponding to this similarity/distance matrix, let us consider two GNNs that produce the embeddings in Table 1. As can be seen, while GNN-1 preserves an identical distance of 2 in the output space among all

Table 1: Embeddings of two GNNs.

| Node | Embeddings from GNN-1 | Embeddings from GNN-2 |
|------|----------------------|----------------------|
| A | $[10, 10, 10, 10]$ | $[10, 10, 10, 10]$ |
| B | $[1, 0, 0, 0]$ | $[1, 0, 0, 0]$ |
| C | $[0, 1, 0, 0]$ | $[0, 2, 0, 0]$ |
| D | $[0, 0, 1, 0]$ | $[0, 0, 2, 0]$ |
| E | $[0, 0, 0, 1]$ | $[0, 0, 0, 2]$ |

nodes except those involving $A$, the inequality is higher in GNN-2. Despite this, the Lipschitz constants for both GNN-1 and GNN-2 are dominated by the $(A, B)$ pair leading to an identical value of 19.5.

We address this limitation with the Gini coefficient.

### 2.1.2 Gini Coefficient

In economics theory, the *Lorenz* curve plots the cumulative share of total income held by the cumulative percentage of individuals ranked by their income, from the poorest to the richest (Gastwirth, 1972; Sitthiyot & Holasut, 2021; Dagum, 1980). The line at 45 degrees thus represents perfect equality of incomes. The *Gini coefficient* is the ratio of the area that lies between the *line of equality* and the Lorenz curve over the total area under the line of equality (Gastwirth, 1972; Sitthiyot & Holasut, 2021) (See Fig. D in Appendix for an example). Mathematically,

$$Gini(\mathcal{X}) = \frac{\sum_{i=1}^{n} \sum_{j=1}^{n} |x_i - x_j|}{2n \sum_{j=1}^{n} x_j}, \tag{2}$$

where $\mathcal{X}$ is the set of individuals, $x_i$ is the income of person $i$ and $n = |\mathcal{X}|$. A lower Gini indicates fairer distribution, with 0 indicating perfect equality.

In our context, rather than measuring income inequality, we assess fairness in algorithmic outcomes by ensuring that similar individuals receive similar predictions (Kang et al., 2020). The analogy to income distribution holds because, just as the Gini coefficient captures disparities across the entire population, our formulation captures disparities in algorithmic outcomes across all pairs of individuals. However, instead of treating all differences equally, we introduce a similarity-weighted extension.

**Definition 6** (Gini Coefficient for Individual fairness)**.** *The individual fairness of the node set $\mathcal{V}$ with embeddings $\mathbf{Z}$ and similarity matrix $\mathbf{S}$ is its weighted Gini coefficient, which is defined as follows,*

$$Gini(\mathcal{V}) = \frac{\sum_{i=1}^{n} \sum_{j=1}^{n} \mathbf{S}[i,j] \|\mathbf{z}_i - \mathbf{z}_j\|_1}{2n \sum_{i=1}^{n} \|\mathbf{z}_i\|_1}. \tag{3}$$

In this formulation, large disparities between dissimilar nodes have minimal impact on the Gini coefficient, while considerable disparities among similar nodes have a stronger effect. The denominator normalizes

disparities by the total weighted sum of outcomes. This prevents the metric from being arbitrarily sensitive to scale and ensures a consistent interpretation across different datasets and applications.[1]

Further, the Gini coefficient is superior to the Lipschitz constant because it is not limited to reflecting only the worst-case scenario. In the example presented in § 2.1.1, for instance, Gini yields values of 0.377 and 0.74 for GNN-1 and GNN-2, respectively (c.f. Table 1), accurately reflecting the higher inequality present in GNN-2's outcomes. Also, it important to highlight the Definition 6, formalizes the multidimensional Gini index to vector-valued node embeddings, a detailed derivation is presented in in Appendix K.

## 2.2 Group Fairness

Optimizing individual fairness alone may cause group disparity (Kang et al., 2020). Specifically, one group may have a considerably higher level of individual fairness than other groups. We follow the same definition of group fairness proposed in GUIDE Song et al. (2022), with the only difference of replacing the Lipschitz constant with the Gini coefficient.

**Definition 7** (Group Fairness)**.** *Group Fairness is satisfied if the levels of individual fairness across all groups are equal. Mathematically, let $\{\mathcal{V}_1, \cdots, \mathcal{V}_m\}$ be the partition of node set $\mathcal{V}$ induced by the sensitive attribute. Group fairness demands $\forall i, j, \ Gini(\mathcal{V}_i) = Gini(\mathcal{V}_j)$.*

To convert group fairness satisfaction into an optimization problem, we introduce the notion of *group disparity of individual fairness*.

**Definition 8** (Group Disparity)**.** *Given a pair of groups $\mathcal{V}_g, \mathcal{V}_h$, the disparity among these groups is quantified as:*

$$GDIF(\mathcal{V}_g, \mathcal{V}_h) = \max\left\{\frac{Gini(\mathcal{V}_g)}{Gini(\mathcal{V}_h)}, \frac{Gini(\mathcal{V}_h)}{Gini(\mathcal{V}_g)}\right\} \tag{4}$$

$GDIF(\mathcal{V}_g, \mathcal{V}_h) \geq= 1$, with a value of 1 indicating perfect satisfaction. Now, to generalize across $m$ groups, *average group disparity* measures the average disparity across all pairs.

$$A\text{-}GDIF(\{\mathcal{V}_1, \cdots, \mathcal{V}_m\}) = \frac{1}{m(m-1)} \sum_{\forall i,j \in [1,m], \ i \neq j} GDIF(\mathcal{V}_i, \mathcal{V}_j) \tag{5}$$

We now argue that minimizing the Gini Coefficient promotes *equal opportunity*.

**Definition 9** (Equal Opportunity Hardt et al. (2016))**.** *The GNN prediction for a node $v$, $\hat{Y}(v)$, satisfies *equal opportunity* if:*

$$\left|P\{\hat{Y}(v) = 1 \mid Y(v) = 1, v \in \mathcal{V}_1\} - P\{\hat{Y}(v) = 1 \mid Y(v) = 1, v \in \mathcal{V}_2\}\right| \leq \varepsilon$$

*for some small $\epsilon$, i.e., we require that the rate of positive outcomes is similar among those who deserve positive outcomes.*

**Observation 1.** *Minimizing the Gini promotes equal opportunity.*

The numerator of Gini (Eq. 3) is $\sum_{i=1}^n \sum_{j=1}^n \mathbf{S}[i,j]\|\mathbf{z}_i - \mathbf{z}_j\|_1$. Since the denominator is independent of the pair-wise distances between node embeddings, it implies:

$$\frac{\partial Gini(\mathcal{V})}{\partial \|\mathbf{z}_i - \mathbf{z}_j\|_1} \propto \mathbf{S}[i,j] \tag{6}$$

Therefore, minimizing $Gini(\mathcal{V})$ will minimize $\|\mathbf{z}_i - \mathbf{z}_j\|_1$ when $\mathbf{S}[i,j]$ is large, which implies that similar nodes will have similar embeddings (small $\|\mathbf{z}_i - \mathbf{z}_j\|_1$). Since GNN outcomes are functions of the embeddings, denoted by $f(\mathbf{z}_i)$, if two nodes $v_i$ and $v_j$ are such that $\|\mathbf{z}_i - \mathbf{z}_j\|_1$ is small, then by the Cauchy-Schwarz inequality,

$$\|f(\mathbf{z}_i) - f(\mathbf{z}_j)\| \approx \|\nabla f(\mathbf{z}_i) \cdot (\mathbf{z}_i - \mathbf{z}_j)\| \leq \|\nabla f(\mathbf{z}_i)\| \cdot \|\mathbf{z}_i - \mathbf{z}_j\|_1,$$

---

[1]Here, the definition is expressed using the $\ell_1$-norm; alternative choices such as the $\ell_2$-norm are possible but correspond to different interpretations of fairness.

i.e., the GNN will yield similar outcomes for reasonably smooth $f$.

From our definition of group fairness (Def. 7) and the corresponding Group Disparity (GDIF) (Def. 8), minimization implies that the internal consistency of embeddings within each group becomes similar. Since similar embeddings yield similar outcomes, the predicted label will be similar for individuals with a true label of 1 within a group. Let $v_1 \in \mathcal{V}_1$ and $v_2 \in \mathcal{V}_2$ be two similar nodes such that $Y(v_1) = Y(v_2) = 1$. Then,

$$\|\mathbf{z}_1 - \mathbf{z}_2\|_1 \leq \epsilon \Rightarrow |P(\hat{Y}(v_1) = 1) - P(\hat{Y}(v_2) = 1)| \leq \epsilon$$

for some small $\epsilon$. For all such similar pairs in $\mathcal{V}_1$ and $\mathcal{V}_2$, we have

$$\left| P\{\hat{Y}(v) = 1 \mid Y(v) = 1, v \in \mathcal{V}_1\} - P\{\hat{Y}(v) = 1 \mid Y(v) = 1, v \in \mathcal{V}_2\} \right| \leq \varepsilon$$

which is the definition of Equal Opportunity.

### 2.3 Problem Formulation

We finally state our problem objective.

**Problem 1** (Fair GNN). *Given a graph $G = (\mathcal{V}, \mathcal{E}, \mathbf{X})$, a symmetric similarity matrix $\mathbf{S}$ for nodes in $\mathcal{V}$, and $k$ disjoint groups differing in their sensitive attributes (i.e, $\cup_{i=1}^{k} \mathcal{V}_i$), our goal is to learn node embeddings $\mathbf{Z}$ such that:*

1. *Overall individual fairness level is maximized (Eq. 3);*

2. *Cumulative group disparity is minimized (Eq. 5).*

3. *The prediction quality on the node embeddings is maximized.*

## 3 GraphGini: Proposed Methodology

The key challenge in learning fair node embeddings $\mathbf{Z}$ is to design a loss function that encapsulates all three objectives of Prob. 1. Designing this loss function requires us to navigate through three key challenges. **(1)** Gini is non-differentiable and hence cannot be integrated directly into the loss function. **(2)** Second, since group disparity is a formulation over a max operator, it remains non-differentiable as well. **(3)** Finally, we need a mechanism to automatically balance the three optimization objectives of the GNN without resorting to manual adjustments of weight parameters.

### 3.1 Optimizing Individual Fairness

The weighted Gini coefficient $Gini(\mathcal{V})$ is not differentiable in general. In case $\|\mathbf{z}_i - \mathbf{z}_j\|_1$ is 0, the derivatives from different directions may converge to different limits. To work around this problem, we present a differentiable (and convex) upper bound to this metric.

**Proposition 1.** *Given node embeddings $\mathbf{Z} \in \mathbb{R}^{n \times c}$ of graph $G = (\mathcal{V}, \mathcal{E}, \mathbf{X})$ with node similarity matrix $\mathbf{S}$ and corresponding Laplacian $\mathbf{L}$ (Recall Def. 4),*

$$Gini(\mathcal{V}) \leq Tr\left(\mathbf{Z}^T \mathbf{L} \mathbf{Z}\right).$$

*Proof.* The $\ell_2$-norm, when multiplied by the square root of the dimension of the space, is an upper bound on the $\ell_1$-norm. Using this fact, we get:

$$\sum_{i=1}^{n} \sum_{j=1}^{n} \mathbf{S}[i,j] \|\mathbf{z}_i - \mathbf{z}_j\|_2 \quad \leq \quad \sum_{j=1}^{|\mathcal{E}|} \mathbf{S}[i,j] \|\mathbf{z}_i - \mathbf{z}_j\|_1 \quad \leq \quad \sqrt{c} \sum_{i=1}^{n} \sum_{j=1}^{n} \mathbf{S}[i,j] \|\mathbf{z}_i - \mathbf{z}_j\|_2 \tag{7}$$

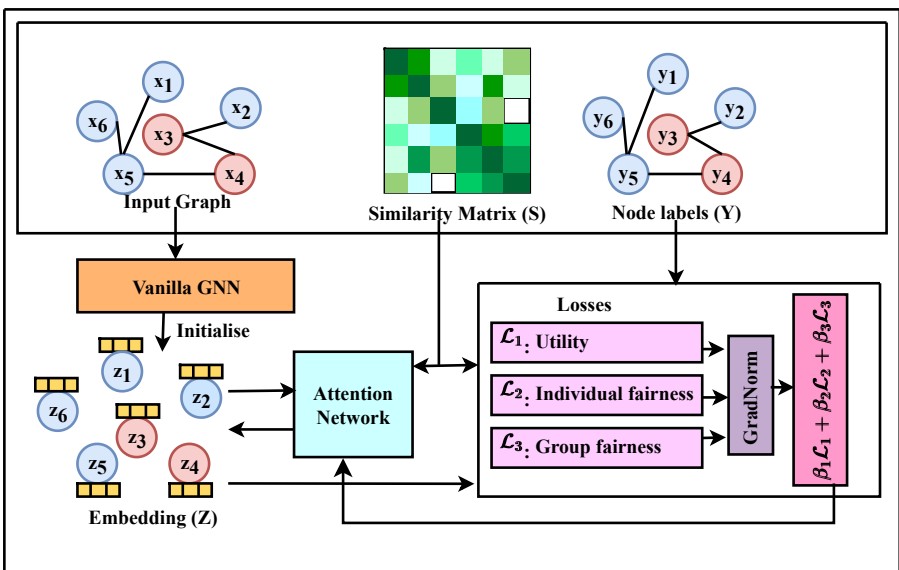

Figure 1: The figure illustrates the pipeline of GRAPHGINI. The sequence of actions depicted in this figure is formally encapsulated in Alg. 1 (in Appendix) and discussed in § 3.3.

where $c$ is the dimension of the GNN embeddings. Now, substituting this inequality into the expression for $Gini(\mathcal{V})$ (Eq. 3), we have:

$$\frac{1}{2\sqrt{c}}\sum_{i=1}^{n}\sum_{j=1}^{n}\mathbf{S}[i,j]\|\mathbf{z}_i - \mathbf{z}_j\|_2 \quad \leq \quad Gini(\mathcal{V}) \quad \leq \quad \frac{1}{2}\sum_{i=1}^{n}\sum_{j=1}^{n}\mathbf{S}[i,j]\|\mathbf{z}_i - \mathbf{z}_j\|_2^2 \; = \; Tr\left(\mathbf{Z}^T\mathbf{L}\mathbf{Z}\right) \quad (8)$$

as the denominator in Eq. 3 is strictly positive and scales both sides of the inequality, thereby does not affect the inequality. $\qquad\square$

The upper bound in Proposition 1 has appeared as a heuristic individual fairness regularizer in prior work Kang et al. (2020). Here, we present a new interpretation based on the Gini coefficient. Hereon, we use the notation $\widehat{Gini}(\mathcal{V}) = Tr\left(\mathbf{Z}^T\mathbf{L}\mathbf{Z}\right)$ to denote the upper bound on Gini with respect to node set $\mathcal{V}$ characterized by similarity matrix $\mathbf{S}$ and embeddings $\mathbf{Z}$. An alternative solution to work around the non-differentiability is to approximate it using smooth pooling-operator surrogates such as softmax and top-$k$ as individual-fairness regularizers. In Appendix K, we demonstrate that these surrogates are empirically inferior to the proposed differentiable upper bound. Next, we discuss the effectiveness of minimizing this differentiable upper bound as a surrogate for the true Gini coefficient. The following proposition outlines how optimisation over the relaxed objective remains aligned with reductions in the original fairness measure, i.e., the Gini coefficient.

**Proposition 2.** *The same value of* $\mathbf{Z}$ *minimizes both* $Gini(\mathcal{V})$ *and its differentiable upper bound* $\widehat{Gini}(\mathcal{V})$. *Moreover, since* $\widehat{Gini}(\mathcal{V}) = \text{Tr}(\mathbf{Z}^T\mathbf{L}\mathbf{Z})$ *is convex in* $\mathbf{Z}$, *we can efficiently find its global minimum. Consequently, solving for the minimizer of* $\widehat{Gini}(\mathcal{V})$ *guarantees minimization of the true Gini coefficient as well.*

*Proof.* The detailed proof is provided in App. E. $\qquad\square$

## 3.2 Optimizing Group Fairness through Nash Social Welfare Program

Group fairness optimization, defined in Eq. 5, is a function of the Gini coefficient. Hence, Eq. 5 is non-differentiable as well. The natural relaxation to obtain a differentiable proxy function is therefore to replace

$Gini(\mathcal{V}_g)$ with $\widehat{Gini}(\mathcal{V}_g) = Tr\left(\mathbf{Z}_g^T \mathbf{L_g} \mathbf{Z}_g\right)$ in Eq. 4. Here $\mathcal{V}_g \subseteq \mathcal{V}$ is a group of nodes, $\mathbf{Z}_g = \{\mathbf{z}_i \in \mathbf{Z} \mid v_i \in \mathcal{V}_g\}$ are the node embeddings of $\mathcal{V}_g$, and $\mathbf{L}_g$ is the Laplacian of the similarity matrix $\mathbf{S}_g : \mathcal{V}_g \times \mathcal{V}_g \to \mathbb{R}$. Unfortunately, even with this modification, optimization of group fairness remains non-differentiable since a function of the form $max\{a,b\}$ is differentiable everywhere except at $a = b$.[2] To circumvent this non-differentiability, we employ *Nash Social Welfare Program (NSWP)* optimization (Charkhgard et al., 2022).

The NSWP is a technique in mathematical programming that combines multiple objectives into a single objective, leading to a Pareto-optimal solution. The core idea involves creating a product of the differences between two minimization terms, both referenced to a common point. In our context, $\frac{\widehat{Gini}(\mathcal{V}_g)}{\widehat{Gini}(\mathcal{V}_h)}$ and $\frac{\widehat{Gini}(\mathcal{V}_h)}{\widehat{Gini}(\mathcal{V}_g)}$ are the competing terms to be minimized. We choose 1 as the common reference. Then, using the NSWP formulation, the resulting optimization leads to the following expression:

$$-\left(\frac{\widehat{Gini}(\mathcal{V}_g)}{\widehat{Gini}(\mathcal{V}_h)} - 1\right)\left(\frac{\widehat{Gini}(\mathcal{V}_h)}{\widehat{Gini}(\mathcal{V}_g)} - 1\right) \tag{9}$$

Intuitively, group fairness is maximized when for any pair of groups $Gini(\mathcal{V}_g) = Gini(\mathcal{V}_h)$. On the other hand, if $Gini(\mathcal{V}_g) \gg Gini(\mathcal{V}_h)$ then Eq. 9 approaches $\infty$. We now establish that the solution obtained by minimizing Eq. 9 is *Pareto optimal*.

**Proposition 3.** *Minimizing Eq. 9 produces Pareto optimal solutions for group fairness over all groups.*

*Proof.* The detailed proof is provided in App. F. □

### 3.3 Learning Framework

GRAPHGINI strives to achieve balance among three distinct goals:

1. **Utility Loss:** We assume $\mathcal{L}_1$ denotes the loss term for the GNN prediction task.

2. **Loss for individual fairness:** To maximize the overall individual fairness level, $Gini(\mathcal{V})$, we minimize:
$$\mathcal{L}_2 = -Tr\left(\mathbf{Z}^T \mathbf{L} \mathbf{Z}\right), \tag{10}$$

3. **Loss for group fairness:** To maximize group fairness across a set of disjoint node partitions $\{V_1, \cdots, \mathcal{V}_m\}$, we minimize:

$$\mathcal{L}_3 = -\frac{1}{m(m-1)} \sum_{\forall i,j \in [1,m],\; i \neq j} \left(\frac{\widehat{Gini}(\mathcal{V}_i)}{\widehat{Gini}(\mathcal{V}_j)} - 1\right)\left(\frac{\widehat{Gini}(\mathcal{V}_j)}{\widehat{Gini}(\mathcal{V}_i)} - 1\right)$$

Incorporating all of the above loss terms, we obtain:

$$\mathcal{L} = \beta_1 \mathcal{L}_1 + \beta_2 \mathcal{L}_2 + \beta_3 \mathcal{L}_3. \tag{11}$$

Here, $\beta_i$s are tunable hyper-parameters.

The overall framework of GRAPHGINI is presented in Fig. 1. Alg. 1 in the Appendix presents the pseudocode. First, embeddings $\mathbf{Z}$ for nodes are learned on the input GNN using only the utility loss. These initial embeddings are now enhanced by injecting fairness through a Graph Attention Network (GAT) (Velickovic et al., 2017), where the attention is reinforced in proportion to the similarity strength.

$$\alpha_{i,j} = \frac{\exp(\text{LeakyReLU}(\mathbf{a}^T[\mathbf{W}\mathbf{h}_i^\ell \| \mathbf{W}\mathbf{h}_j^\ell])\mathbf{S}[i,j])}{\sum_{j \in \mathcal{N}_i} \exp(\text{LeakyReLU}(\mathbf{a}^T[\mathbf{W}\mathbf{h}_i^\ell \| \mathbf{W}\mathbf{h}_j^\ell])\mathbf{S}[i,j])}, \tag{12}$$

Here, $\mathbf{W} \in \mathbb{R}^{d \times h}$ is the weight matrix, $\mathbf{h}^\ell \in \mathbb{R}^h$ is the embedding for node $i$ in layer $\ell$, $\mathbf{a} \in \mathbb{R}^{2d}$ is the attention vector. The symbol $[.\|.]$ denotes concatenation of vectors and $\mathcal{N}_i$ represents the neighbourhood of node $i$. Using pairwise attention, the embeddings are aggregated as $\mathbf{h}_i^{\ell+1} = \sigma(\Sigma_{j \in \mathcal{N}_i} \alpha_{i,j} \mathbf{W}\mathbf{h}_j^\ell)$.

---

[2]While ReLU is also a max operator, it represents a special case handled via pre-coded values. They do not transfer to generic loss functions. See App. E.1 for details.

Table 2: Summary statistics of used datasets.

| Name | # nodes | # node attributes | # of edges in $A$ | # edges in $S$ | Sensitive Attribute |
|---|---|---|---|---|---|
| Credit | 30,000 | 13 | 304,754 | 1, 687, 444 | age |
| Income | 14,821 | 14 | 100,483 | 1, 997, 641 | race |
| Pokec-n | 66,569 | 266 | 1,100,663 | 32, 837, 463 | age |

### 3.4 Balanced Optimization with GradNorm

The optimisation process requires a delicate calibration of weighing factors, i.e., coefficients $\beta_i$'s in our loss function (Eq. 11). Manual tuning, the predominant approach among existing techniques, is challenging as these weights not only determine the relative importance of fairness objectives compared to utility but also play a dual role in normalizing individual terms within the loss function. We perform *gradient normalization* to automatically learn the weights (Chen et al., 2018).

Let $\theta_t$ be the parameters of the GAT at epoch $t$ and $\mathcal{L}_i(t)$, $i \in \{1, 2, 3\}$ denotes the value of the loss term $\mathcal{L}_i$ in the $t$-th epoch. We initiate the learning process with initialized weights $\beta_i(0)$. In later rounds ($t > 0$), we treat each $\beta_i(t)$ as adjustable parameters aimed at minimizing a modified loss function, which we will develop next.

First, corresponding to each $\mathcal{L}_i(t)$, we compute the $\ell_2$ norm of the gradient.

$$G_i(t) = \|\Delta_{\theta_t} \beta_i(t)\mathcal{L}_i(t)\|_2 \tag{13}$$

The *loss ratio* in the $t$-th epoch is measured as $R_i(t) = \frac{\mathcal{L}_i(t)}{\mathcal{L}_i(0)}$, which reveals the inverse *training rate*, i.e., the lower the ratio, the higher is the training. Finally, instead of optimizing the original loss in $\mathcal{L}$ ( Eq. 11), we minimize the loss below that operates on the gradient space of the individual loss terms. Specifically,

$$\mathcal{L}_{gradnorm}(t) = \sum_{i=1}^{3} \left\| \beta_i(t)G_i(t) - \bar{G}(t) \times \frac{R_i(t)}{\bar{R}(t)} \right\|_1 \tag{14}$$

$$\text{where,} \quad \bar{G}(t) = \frac{1}{3}\sum_{i=1}^{3} G_i(t), \qquad \bar{R}(t) = \frac{1}{3}\sum_{i=1}^{3} R_i(t)$$

Intuitively, Eq. 14 strives to achieve similar training rates across all individual loss terms. Hence, it penalizes weighing factor $\beta_i(t)$ if the training rate is higher in $\mathcal{L}_i(t)$ than the average across all terms.

### 3.5 Convergence and Time Complexity

**Convergence.** Since our framework uses standard GNN architectures, the convergence analysis of gradient descent for GNNs from Awasthi et al. (2021) which assumes a smooth and bounded loss function, directly applies. Therefore, the optimization procedure for GRAPHGINI is guaranteed to converge under the same conditions. A detailed discussion over convergence is provided in the Appendix G.

**Time complexity.** The base Graph Neural Network (GNN) has a time complexity of $O(|\mathcal{E}|)$, where $|\mathcal{E}|$ represents the number of edges in the graph. Both GRAPHGINI and the primary baseline, GUIDE, require an additional step of computing the Laplacian matrix. This computation is linear with respect to the number of non-zero entries in the similarity matrix, which in the worst-case scenario is $O(|\mathcal{V}|^2)$, where $|\mathcal{V}|$ is the number of vertices in the graph. The overall time complexity of these fair GNN approaches can thus be expressed as $O(|\mathcal{E}| + |\mathcal{V}|^2)$, accounting for both the base GNN operations and the Laplacian computation. The runtime comparison is also empirically demonstrated in Appendix G.

## 4 Experiments

The objective in this section is to answer the following questions:

Table 3: Performance comparison of benchmarked algorithms across datasets using the GCN backbone architecture. "Vanilla" indicates no debiasing. Arrows: ↑ = higher is better, ↓ = lower is better. Best performance per column is in **bold**. Individual fairness (IF) is reported in thousands.

| Model | AUC (↑) | IF (↓) | GD (↓) | IF-Gini (↓) | GD-Gini (↓) |
|---|---|---|---|---|---|
| **Credit** | | | | | |
| Vanilla | 0.68±0.10 | 40.01±2.16 | 1.34±0.05 | 0.30±0.01 | 1.72±0.02 |
| FairGNN | 0.68±0.05 | 23.35±13.39 | 1.34±0.10 | 0.26±0.02 | 1.69±0.01 |
| NIFTY | 0.69±0.01 | 30.85±1.45 | 1.25±0.05 | 0.26±0.02 | 1.67±0.01 |
| PFR | 0.64±0.16 | 36.57±7.90 | 1.46±0.02 | 0.30±0.00 | 1.72±0.01 |
| InFoRM | 0.68±0.02 | 2.42±0.02 | 1.45±0.02 | 0.18±0.01 | 1.73±0.01 |
| PostProcess | **0.70**±0.00 | 40.23±0.00 | 1.40±0.00 | 0.30±0.02 | 1.74±0.01 |
| iFairNMTF | 0.69±0.01 | 40.62±1.34 | 1.38±0.02 | 0.29±0.02 | 1.71±0.01 |
| GNN GEI | 0.69±0.00 | 40.21±1.32 | 1.35±0.05 | 0.28±0.01 | 1.70±0.02 |
| TF-GNN | 0.69±0.00 | 12.80±0.00 | 1.45±0.00 | 0.19±0.00 | 1.71±0.02 |
| BeMAP | 0.68±0.00 | 3.11±1.23 | 1.20±0.00 | 0.15±0.02 | 1.73±0.02 |
| FairSIN | 0.68±0.00 | 8.23±3.20 | 1.15±0.00 | 0.20±0.02 | 1.69±0.01 |
| GUIDE | 0.68±0.02 | 1.94±0.10 | **1.00**±0.00 | 0.14±0.01 | 1.41±0.01 |
| GRAPHGINI | 0.68±0.00 | **0.22**±**0.06** | **1.00**±0.00 | **0.12**±0.01 | **1.10**±0.01 |
| **Income** | | | | | |
| Vanilla | 0.77±0.01 | 370.81±0.02 | 1.30±0.01 | 0.32±0.01 | 1.56±0.02 |
| FairGNN | 0.76±0.00 | 250.22±85.32 | 1.16±0.03 | 0.30±0.02 | 1.55±0.01 |
| NIFTY | 0.73±0.00 | 44.32±5.67 | 1.39±0.03 | 0.25±0.01 | 1.60±0.01 |
| PFR | 0.75±0.01 | 245.95±0.50 | 1.33±0.01 | 0.30±0.02 | 1.56±0.02 |
| InFoRM | **0.78**±0.01 | 199.20±0.04 | 1.35±0.04 | 0.26±0.02 | 1.54±0.00 |
| PostProcess | 0.77±0.00 | 367.62±0.00 | 1.28±0.00 | 0.31±0.01 | 1.55±0.01 |
| iFairNMTF | 0.77±0.00 | 358.20±0.32 | 1.28±0.01 | 0.30±0.02 | 1.57±0.01 |
| GNN GEI | 0.77±0.00 | 357.23±5.04 | 1.47±0.01 | 0.30±0.02 | 1.56±0.01 |
| TF-GNN | 0.76±0.00 | 25.65±0.00 | 1.85±0.00 | 0.24±0.01 | 1.60±0.01 |
| BeMAP | 0.73±0.00 | 211.66±25.34 | 1.33±0.00 | 0.26±0.01 | 1.57±0.01 |
| FairSIN | 0.73±0.00 | 287.22±38.75 | 1.42±0.00 | 0.27±0.01 | 1.61±0.01 |
| GUIDE | 0.73±0.01 | 33.20±12.14 | **1.00**±0.00 | 0.20±0.01 | 1.10±0.01 |
| GRAPHGINI | 0.73±0.09 | **21.12**±**5.22** | **1.00**±0.00 | **0.17**±0.01 | **1.02**±0.01 |
| **Pokec-n** | | | | | |
| Vanilla | **0.77**±0.01 | 950.28±39.11 | 6.93±0.10 | 0.41±0.02 | 1.80±0.01 |
| FairGNN | 0.69±0.03 | 363.73±78.58 | 6.29±1.28 | 0.35±0.01 | 1.76±0.02 |
| NIFTY | 0.74±0.00 | 85.25±10.55 | 5.06±0.29 | 0.31±0.01 | 1.72±0.02 |
| PFR | 0.53±0.00 | 98.25±9.44 | 15.84±0.03 | 0.32±0.02 | 1.70±0.01 |
| InFoRM | **0.77**±0.00 | 230.45±6.13 | 6.62±0.10 | 0.34±0.01 | 1.69±0.01 |
| PostProcess | **0.77**±0.00 | 872.12±82.23 | 5.93±0.27 | 0.39±0.01 | 1.72±0.02 |
| iFairNMTF | 0.76±0.00 | 781.29±98.45 | 7.23±0.11 | 0.39±0.01 | 1.73±0.01 |
| GNN GEI | **0.77**±0.00 | 875.11±9.31 | 6.43±8.31 | 0.37±0.01 | 1.74±0.02 |
| TF-GNN | 0.74±0.00 | 245.48±11.43 | 9.28±0.10 | 0.33±0.02 | 1.81±0.01 |
| BeMAP | 0.73±0.00 | 372.00±48.75 | 2.29±0.00 | 0.31±0.01 | 1.41±0.02 |
| FairSIN | 0.73±0.00 | 482.00±39.20 | 2.88±0.03 | 0.31±0.02 | 1.41±0.03 |
| GUIDE | 0.73±0.02 | 55.05±30.87 | 1.11±0.03 | 0.24±0.02 | 1.19±0.01 |
| GRAPHGINI | 0.74±0.00 | **31.10**±**5.22** | **1.00**±0.00 | **0.21**±0.01 | **1.14**±0.01 |

- **RQ1:** How well GRAPHGINI balances utility, individual fairness, and group fairness objectives compared to baselines?

Table 4: Performance comparison of benchmarked algorithms across datasets using the GIN backbone architecture. "Vanilla" indicates no debiasing. Arrows: ↑ = higher is better, ↓ = lower is better. Best performance per column is in **bold**. Individual fairness (IF) is reported in thousands.

| Model | AUC (↑) | IF (↓) | GD (↓) | IF-Gini (↓) | GD-Gini (↓) |
|---|---|---|---|---|---|
| **Credit** | | | | | |
| Vanilla | **0.71±0.00** | 118.02±16.22 | 1.80±0.16 | 0.29 ± 0.02 | 1.64 ± 0.01 |
| FairGNN | 0.68±0.05 | 76.10±45.22 | 2.20±0.15 | 0.25 ± 0.01 | 1.64 ± 0.01 |
| NIFTY | 0.70±0.04 | 58.33±39.85 | 1.68±0.22 | 0.25 ± 0.01 | 1.64 ± 0.01 |
| PFR | **0.71±0.03** | 160.55±100.20 | 2.44±1.20 | 0.30 ± 0.01 | 1.66 ± 0.01 |
| InFoRM | 0.69±0.03 | 2.96±0.12 | 1.75±0.15 | 0.16 ± 0.01 | 1.66 ± 0.01 |
| PostProcess | **0.71±0.00** | 177.44±0.66 | 1.42±0.08 | 0.30 ± 0.01 | 1.66 ± 0.01 |
| iFairNMTF | **0.71±0.01** | 107.72±9.04 | 1.54±0.23 | 0.28 ± 0.02 | 1.66 ± 0.02 |
| GNN GEI | 0.70±0.06 | 176.21±1.21 | 1.45 ± 0.23 | 0.27 ± 0.02 | 1.65 ± 0.01 |
| TF-GNN | **0.71±0.01** | 18.42±0.42 | 1.29±0.11 | 0.21 ± 0.01 | 1.70 ± 0.01 |
| BeMAP | 0.68±0.00 | 12.25±3.25 | 1.39±0.01 | 0.17 ± 0.00 | 1.70 ± 0.00 |
| FairSIN | 0.68±0.0 | 18.12± 6.68 | 1.60±0.00 | 0.19 ± 0.01 | 1.70 ± 0.01 |
| GUIDE | 0.68±0.02 | 2.45±0.03 | **1.00±0.00** | 0.12 ± 0.01 | 1.30 ± 0.01 |
| GRAPHGINI | 0.68±0.00 | **1.92±0.09** | **1.00±0.00** | **0.10 ± 0.01** | **1.13 ± 0.01** |
| **Income** | | | | | |
| Vanilla | **0.80±0.02** | 2812.62±1061.04 | 1.87±0.46 | 0.46 ± 0.01 | 2.00 ± 0.00 |
| FairGNN | 0.79±0.00 | 1359.93±880.22 | 3.32±1.20 | 0.35 ± 0.01 | 2.06 ± 0.01 |
| NIFTY | 0.79±0.01 | 617.11±320.13 | 1.16±0.30 | 0.29 ± 0.02 | 1.65 ± 0.01 |
| PFR | 0.79±0.00 | 2210.35±461.11 | 2.35±1.14 | 0.45 ± 0.02 | 2.03 ± 0.02 |
| InFoRM | 0.80±0.01 | 309.35±14.24 | 1.61±0.28 | 0.27 ± 0.01 | 2.05 ± 0.01 |
| PostProcess | 0.80±0.00 | 420.78±128 | 2.5±0.01 | 0.28 ± 0.01 | 2.10 ± 0.02 |
| iFairNMTF | 0.80±0.00 | 2574.38±134.62 | 2.43±0.38 | 0.45 ± 0.02 | 2.10 ± 0.03 |
| GNN GEI | 0.79±0.00 | 2531.59±78.12 | 3.07±0.23 | 0.44 ± 0.03 | 2.13 ± 0.01 |
| TF-GNN | 0.80±0.01 | 310.20±1.20 | 1.28±0.01 | 0.28 ± 0.01 | 1.71 ± 0.02 |
| BeMAP | 0.75±0.00 | 872.89±75.11 | 1.34±0.00 | 0.30 ± 0.01 | 1.71 ± 0.02 |
| FairSIN | 0.75±0.0 | 929.00± 65.80 | 1.45±0.00 | 0.30 ± 0.00 | 1.75 ± 0.01 |
| GUIDE | 0.74±0.01 | 83.85±20.20 | **1.00±0.00** | 0.20 ± 0.01 | 1.14 ± 0.01 |
| GRAPHGINI | 0.74±0.00 | **55.73± 9.12** | **1.00±0.00** | **0.16 ± 0.01** | **1.09 ± 0.01** |
| **Pokec-n** | | | | | |
| Vanilla | **0.76±0.00** | 4490.50±1550.80 | 8.38±1.30 | 0.45 ± 0.02 | 1.81 ± 0.02 |
| FairGNN | 0.69±0.01 | 416.28±402.83 | 4.84±2.94 | 0.36 ± 0.01 | 1.52 ± 0.02 |
| NIFTY | **0.76±0.01** | 2777.36±346.29 | 9.28±0.28 | 0.39 ± 0.02 | 1.80 ± 0.01 |
| PFR | 0.60±0.01 | 628.27±85.89 | 6.20±0.79 | 0.36 ± 0.02 | 1.56 ± 0.01 |
| InFoRM | 0.75±0.01 | 271.65±30.63 | 6.83±1.34 | 0.33 ± 0.01 | 1.55 ± 0.02 |
| PostProcess | 0.75±0.00 | 4261.32±113.88 | 9.76±0.25 | 0.44 ± 0.01 | 1.80 ± 0.01 |
| iFairNMTF | 0.75±0.00 | 3972.55±69.34 | 8.45±0.21 | 0.41 ± 0.01 | 1.80± 0.02 |
| GNN GEI | 0.75±0.01 | 4383.26±319.56 | 7.29±0.87 | 0.44 ± 0.02 | 1.75 ± 0.02 |
| TF-GNN | 0.75±0.00 | 268.32±21.82 | 9.31±1.22 | 0.33 ± 0.01 | 1.79 ± 0.02 |
| BeMAP | 0.75±0.00 | 521.00±72.85 | 3.32±0.23 | 0.30 ± 0.02 | 1.24 ± 0.01 |
| FairSIN | 0.75±0.0 | 556.62± 102.38 | 3.42±0.68 | 0.29 ± 0.01 | 1.25 ± 0.01 |
| GUIDE | 0.74±0.01 | 120.65±17.33 | 1.12±0.03 | 0.25 ± 0.01 | 1.21 ± 0.01 |
| GRAPHGINI | 0.74±0.00 | **85.10±6.29** | **1.00± 0.00** | **0.22 ± 0.02** | **1.15 ± 0.01** |

- **RQ2:** How robust is GRAPHGINI across GNN architectures?

- **RQ3:** Is GRAPHGINI robust across similarity matrix variations?

Table 5: Comparison of GRAPHGINI with and without Gradient Normalization (GRAPHGINI WGN).

| Model | AUC(↑) | IF(↓) | GD(↓) | AUC(↑) | IF(↓) | GD(↓) | AUC(↑) | IF(↓) | GD(↓) |
|---|---|---|---|---|---|---|---|---|---|
| | | GCN | | | GIN | | | JK | |
| Credit | | | | | | | | | |
| GraphGini WGN | 0.68±0.00 | 1.82±0.13 | 1.00±0.00 | 0.68±0.00 | 2.15±0.03 | 1.00±0.00 | 0.68±0.00 | 2.01±0.01 | 1.00±0.00 |
| GraphGini | 0.68±0.00 | 0.22±0.06 | 1.00±0.00 | 0.68±0.00 | 1.92±0.09 | 1.00±0.00 | 0.68±0.00 | 1.88±0.02 | 1.00±0.00 |
| Income | | | | | | | | | |
| GraphGini WGN | 0.73±0.09 | 21.12±5.22 | 1.00±0.00 | 0.74±0.00 | 55.73± 9.12 | 1.00±0.00 | 0.75±0.00 | 31.23± 3.22 | 1.00±0.00 |
| GraphGini | 0.73±0.09 | 20.50±3.10 | 1.00±0.00 | 0.74±0.00 | 49.03± 6.33 | 1.00±0.00 | 0.75±0.00 | 29.47± 4.01 | 1.00±0.00 |
| Pokec-n | | | | | | | | | |
| GraphGini WGN | 0.74±0.00 | 31.10±5.22 | 1.00± 0.00 | 0.74±0.00 | 85.10±6.29 | 1.00± 0.00 | 0.78±0.10 | 44.51±0.72 | 1.00± 0.00 |
| GraphGini | 0.74±0.00 | 27.60±6.32 | 1.00± 0.00 | 0.74±0.00 | 81.37±9.87 | 1.00± 0.00 | 0.78±0.10 | 43.87±2.36 | 1.00± 0.00 |

- **RQ4:** Ablation study– What are the individual impacts of the various components on GRAPHGINI?

Our codebase is available at `https://github.com/idea-iitd/GraphGini`.

## 4.1 Datasets

Table 2 presents a summary of the real-world datasets used for benchmarking GRAPHGINI. **Credit Dataset (Yeh & Lien, 2009):** The graph contains 30,000 individuals, who are connected based on their payment activity. The class label whether an individual defaulted on a loan.
**Income Dataset (Song et al., 2022):** This is a similarity graph over 14,821 individuals sampled from the Adult Data Set (Dua & Graff, 2017). The class label indicates whether an individual's annual income exceeds $50,000.
**Pokec-n (Takac & Zabovsky, 2012):** Pokec-n is a social network where the class label indicates the occupational domain of a user.

## 4.2 Empirical Framework

**Metrics:** We use AUCROC and F1-score to assess performance in node classification. To evaluate individual and group fairness, we use *Gini* as well as the metrics used by GUIDE (Song et al., 2022): individual fairness (IF) = $\mathrm{Tr}(\mathbf{Z}^T\mathbf{L}\mathbf{Z})$ and Group disparity (GD). For two groups $g$ and $h$, $GD = \max\{\epsilon_g/\epsilon_h, \epsilon_h/\epsilon_g\}$) where $\epsilon_g = \mathrm{Tr}(\mathbf{Z}^T\mathbf{L}_g\mathbf{Z})$ and $\epsilon_h = \mathrm{Tr}(\mathbf{Z}^T\mathbf{L}_h\mathbf{Z})$. We also used Gini-based fairness metrics: $IF\text{-}Gini = Gini$ (Def. 6) and $GD\text{-}Gini = \max\{G_g/G_h, G_h/G_g\}$) for individual and group disparity, respectively, where $G_f$ denotes the Gini for group $f$. We report the average GD across all pairs of groups. For group disparity, $GD = 1$ is the ideal case, with higher values indicating poorer performance. We also evaluate *Equal Opportunity (EO)*, a classical measure of group fairness.

**Backbones GNNs:** We evaluate on three distinct GNN backbones: GCN (Kipf & Welling, 2016), GIN, (Xu et al., 2018a), and Jumping Knowledge (JK) (Xu et al., 2018b).

**Baseline methods:** We benchmark against eleven baselines, namely (1) GUIDE (Song et al., 2022), (2) FairGNN, (Dai & Wang, 2021), (3) NIFTY (Agarwal et al., 2021), (4) PFR (Lahoti et al., 2020), (5) InFoRM (Kang et al., 2020), (6) PostProcess(Lohia et al., 2019), (7) GEI (Speicher et al., 2018), (8) iFairNMTF (Ghodsi et al., 2024), (9) TF-GNN (Song et al., 2023), (10) BeMap (Lin et al., 2024), and (11) FairSIN (Yang et al., 2024). GUIDE is the state of the art in this space. A more detailed summary of each of the baseline algorithms is provided in App. D. The reproducibility details for baselines, hyperparameter settings, and implementation specifications are given in the Appendix H.

**Similarity matrix:** We evaluate on two different settings: **(1)** *topological similarity*, and **(2)** *attribute similarity*. To instantiate $\mathbf{S}$ for topological similarity, the $(i,j)$-th entry in $\mathbf{S}$ represents the cosine similarity between the $i$-th row and the $j$-th row of the adjacency matrix $\mathbf{A}$. This is aligned with the similarity metric for evaluating fairness in existing works (Kang et al., 2020). For the setting of attribute similarity, the $(i,j)$-th entry in $\mathbf{S}$ represents the cosine between the attributes of $v_i$ and $v_j$ after masking out the sensitive attributes.

Table 6: Impact of fairness regularizer, removing individual fairness constraint, i.e $\beta_2 = 0$.

| Model | GCN | | | GIN | | | JK | | |
|---|---|---|---|---|---|---|---|---|---|
| | AUC(↑) | IF(↓) | GD(↓) | AUC(↑) | IF(↓) | GD(↓) | **AUC(↑)** | **IF(↓)** | **GD(↓)** |
| Credit | | | | | | | | | |
| **Guide** | **0.68** | 16.52 | **1.00** | **0.68** | 17.46 | **1.00** | **0.68** | 9.38 | **1.00** |
| **GraphGini** | **0.68** | **13.77** | **1.00** | **0.68** | **13.35** | **1.00** | **0.68** | **9.28** | **1.00** |
| Income | | | | | | | | | |
| **Guide** | **0.73** | 26.07 | **1.00** | **0.74** | 348.69 | **1.00** | **0.74** | 73.04 | 1.00 |
| **GraphGini** | **0.73** | **25.78** | **1.00** | **0.74** | **184.81** | **1.00** | **0.74** | **66.12** | **1.00** |
| Pokec-n | | | | | | | | | |
| **Guide** | **0.74** | 39.57 | **1.00** | **0.74** | 86.23 | **1.00** | **0.75** | 62.80 | **1.00** |
| **GraphGini** | **0.74** | **33.42** | **1.00** | **0.74** | **38.00** | **1.00** | **0.76** | **41.67** | **1.00** |

Table 7: Impact of fairness regularizer, removing group fairness constraint, i.e $\beta_3 = 0$.

| Model | GCN | | | GIN | | | JK | | |
|---|---|---|---|---|---|---|---|---|---|
| | AUC(↑) | IF(↓) | GD(↓) | AUC(↑) | IF(↓) | GD(↓) | AUC(↑) | IF(↓) | GD(↓) |
| Credit | | | | | | | | | |
| **Guide** | **0.68** | 2.16 | **1.52** | **0.68** | 2.40 | 1.46 | **0.68** | 2.67 | 1.82 |
| **GraphGini** | **0.68** | **2.11** | **1.52** | **0.68** | **2.37** | **1.45** | **0.68** | **2.56** | **1.52** |
| Income | | | | | | | | | |
| **Guide** | **0.72** | 25.38 | **1.02** | 0.74 | 152.35 | 1.23 | **0.74** | 439.27 | 1.14 |
| **GraphGini** | **0.72** | **21.35** | 1.09 | **0.75** | **150.26** | **1.15** | **0.74** | **402.27** | **1.02** |
| Pokec-n | | | | | | | | | |
| **Guide** | **0.74** | 36.34 | 1.21 | **0.74** | 91.47 | 1.56 | **0.75** | 51.46 | 1.32 |
| **GraphGini** | **0.74** | **22.14** | **1.11** | **0.74** | **87.27** | **1.41** | **0.76** | **46.9** | **1.22** |

### 4.3 RQ1 and RQ2: Efficacy of GraphGini and Robustness to Architectures

Table 3 and 4 present a comprehensive evaluation of GRAPHGINI against state-of-the-art baselines encompassing all three datasets, established metrics in the literature and three distinct GNN architectures (results for JK are presented Table M in the Appendix). In this experiment, the similarity matrix is based on topological similarity. A clear trend emerges. While GRAPHGINI suffers a minor decrease in AUCROC when compared to the vanilla backbone GNN, it comprehensively surpasses all baselines in individual and group fairness. Compared to GUIDE, which is the most recent and the only work to consider both individual and group fairness, GRAPHGINI outperforms it across all metrics and datasets. Specifically, GRAPHGINI is never worse in AUCROC, while always ensuring a higher level of individual and group fairness. In terms of numbers, for the Credit dataset, GRAPHGINI improves individual fairness by 88%, 11%, and 14% as compared to GUIDE when embeddings are initialised by GCN, GIN and JK backbone architectures, respectively. Similarly, we observe significant improvement in individual fairness for the Income dataset by 36%, 33%, & 26% and 43%, 29 %, & 46 % for the Pokec-n dataset without hurting on utility and maintaining group fairness. In terms of Gini-based individual (IF-Gini) and group fairness (GD-Gini) metrics, GRAPHGINI outperforms across all GNN architectures and datasets. While this is not surprising since we explicitly optimize for Gini, we note that this also leads to superior performance in the IF and GD measures when computed with the loss function used by GUIDE. This highlights that when the metric optimizes over the entire spectrum of outcomes, rather than just the worst-case scenario, GRAPHGINI produces fairer predictions. Beyond GUIDE,

we note that several of the baselines only optimize group fairness. Yet, GRAPHGINI outperforms all of them in this metric while also optimizing individual fairness.

Table 8: Equal Opportunity (EO) comparisons for GRAPHGINI and baselines on three datasets. ↓ indicates the smaller the value is, the better. Best performances are in bold.

| Model | Credit | Income | Pokec-n |
|---|---|---|---|
| Vanilla GCN | $13.92 \pm 6.00$ | $14.21 \pm 0.15$ | $3.17 \pm 1.10$ |
| PRF | $13.96 \pm 0.79$ | $12.15 \pm 0.03$ | $1.82 \pm 0.18$ |
| FairGNN | $13.77 \pm 0.91$ | $12.01 \pm 0.03$ | $1.85 \pm 0.11$ |
| InFoRM | $14.82 \pm 4.18$ | $11.44 \pm 0.52$ | $3.58 \pm 1.15$ |
| NIFTY | $13.07 \pm 0.63$ | $14.22 \pm 0.58$ | $7.32 \pm 0.94$ |
| GUIDE | $13.54 \pm 0.06$ | $12.35 \pm 0.15$ | $0.80 \pm 0.18$ |
| PostProcess | $13.82 \pm 0.01$ | $13.21 \pm 0.11$ | $1.20 \pm 0.10$ |
| iFairNMTF | $15.13 \pm 0.03$ | $14.11 \pm 0.10$ | $2.15 \pm 0.20$ |
| GNN GEI | $15.20 \pm 0.12$ | $14.20 \pm 0.25$ | $4.60 \pm 0.42$ |
| TF-GNN | $14.62 \pm 0.32$ | $13.30 \pm 0.18$ | $3.85 \pm 0.25$ |
| BeMAP | $15.40 \pm 0.32$ | $14.45 \pm 0.15$ | $4.65 \pm 0.10$ |
| FairSIN | $14.82 \pm 0.64$ | $12.22 \pm 0.50$ | $3.50 \pm 0.52$ |
| GraphGini | $\mathbf{12.20 \pm 0.03}$ | $\mathbf{9.75 \pm 0.25}$ | $\mathbf{0.80 \pm 0.08}$ |

### 4.4 Impact on Equal Opportunity (EO)

The results in Table 8 support Observation 1, showing that GRAPHGINI improves on the classical group fairness measure, Equal Opportunity (EO). We see that GRAPHGINI beats all baseline methods across the datasets on this measure. Overall, this experiment shows that our method is able to connect the study of fairness in GNNs to the Economics literature by optimizing a metric that is considered important in that discipline.

### 4.5 RQ3: Robustness to Similarity Matrix

In the next experiment, we use similarity matrix based on attribute similarity. In this case, the groups are created through $k$-means. The number of clusters are selected based on elbow plot (See Fig. E in the Appendix for details). The primary objectives in this experiment are threefold. Does GRAPHGINI continue to outperform GUIDE, the primary baseline, when similarity is on attributes? How well do these algorithms perform on the metric of Gini coefficient? How is the Gini coefficient distributed across groups (clusters)?

The findings are summarized in Table L (in Appendix) on all three datasets for three GNN architectures mirroring the trends observed in Table 3. GRAPHGINI consistently exhibits the best balance across all three metrics and outperforms GUIDE across all datasets and architectures on average. Additionally, we delve into the Gini coefficient analysis for each group (designated as Cl$X$). As evident from Table L (in Appendix), GRAPHGINI achieves lower Gini coefficients across most clusters, underscoring the effectiveness of the regularizers.

### 4.6 Utility-Fairness Curves

We compare the full utility-fairness curves for GRAPHGINI and baseline models across all three datasets in Figure 3. Here, utility is measured using AUC. For each baseline, we identified the point where its AUC matches that of GRAPHGINI and compared the corresponding Fairness values. For baselines that could not reach the AUC of GRAPHGINI, the IF was measured at their best achievable performance. Across all datasets, GRAPHGINI consistently demonstrates higher fairness at comparable utility, confirming its effectiveness in achieving a favourable trade-off between predictive performance and fairness.

### 4.7 RQ4: Ablation Study

**Impact of Gradient Normalization:** Table 5 presents the performance of GRAPHGINI with manual tuning (denoted as GRAPHGINI WGN) against automatic tuning through gradient normalization where all weights are initialized to 1. Gradient normalization imparts significant improvement in individual performance while achieving the same quality in group fairness and accuracy. Individual fairness (IF) benefits the most since the regularizer term corresponding to IF in our loss function is of the smallest magnitude. Hence, when set to equal weights, IF gets dominated by the other two terms in the loss. With gradient normalization, this issue is circumvented. Finally, it is noteworthy that even without gradient normalization, GRAPHGINI-WGN outperforms GUIDE (refer to Table 3) across all datasets. This underscores that while gradient normalization contributes to improvement, it is not the sole reason for the superiority of GRAPHGINI over GUIDE. More detailed insights into the evolution of the automatically-tuned weight parameters via gradient normalization are provided in App. J.

**Impact of attention:** We conduct an ablation study comparing results with and without attention. Our findings show that attention primarily improves individual fairness. Fig. 2 illustrates these results on individual fairness. This observation is expected since attention is weighted based on similarity to neighbors, promoting nodes to prioritize similar neighbors in their embeddings. Consequently, individual fairness, which advocates for similar individuals receiving similar outcomes, is reinforced.

**Impact of regularizers:** Next, we study the impact of the regularizers corresponding to individual and group fairness on the performance of GRAPHGINI as well as GUIDE. To turn off a particular regularizer, we fix its weight to 0. Table 6 and Table 7 presents the results. Three key observations emerge. Firstly, as anticipated, both individual fairness and group fairness suffer when their respective regularizers are deactivated (compare the metrics of GRAPHGINI and GUIDE in Table 7 with Table 3). Second, while group fairness remains unaffected from turning off individual fairness, the reverse is not true. This phenomenon occurs since group fairness (Eq. 4) is a function over individual fairness. Thus, even when individual fairness is not directly optimized, it gets indirect assistance from optimizing group fairness. Finally, GRAPHGINI maintains its superiority over GUIDE, even with specific regularizers turned off. A more granular trade-off between utility, individual fairness, and group fairness for GRAPHGINI is provided in Appendix M. The results clearly show that significant improvements in fairness metrics can be achieved with minimal impact on accuracy. For example, by compromising 2% in accuracy, we can achieve a 90% increase in individual fairness and a 30% increase in group fairness. We additionally evaluate the robustness of GRAPHGINI under variations in graph homophily/heterophily, attribute noise levels, group imbalance ratios, and hidden embedding dimensions. A detailed ablation analysis covering these factors is provided in Appendices N, O, P, and Q.

### 4.8 Running Time

Table 9 compares the running times of GRAPHGINI with GUIDE, both of which are similar. The vanilla model is faster since it does not account for the similarity matrix, which GRAPHGINI and GUIDE need to

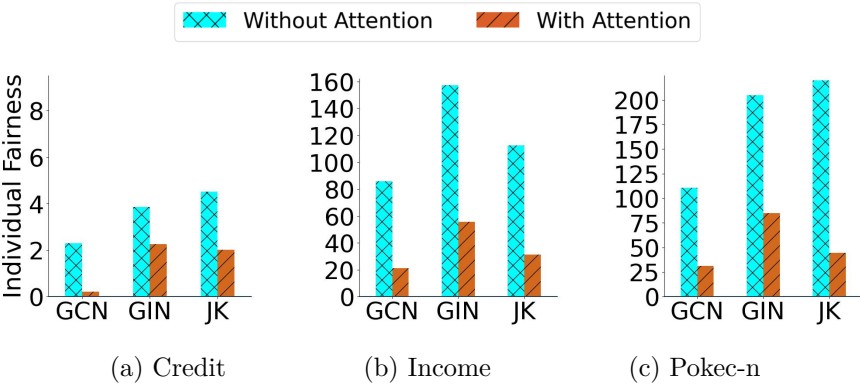

Figure 2: Impact of attention on individual fairness.

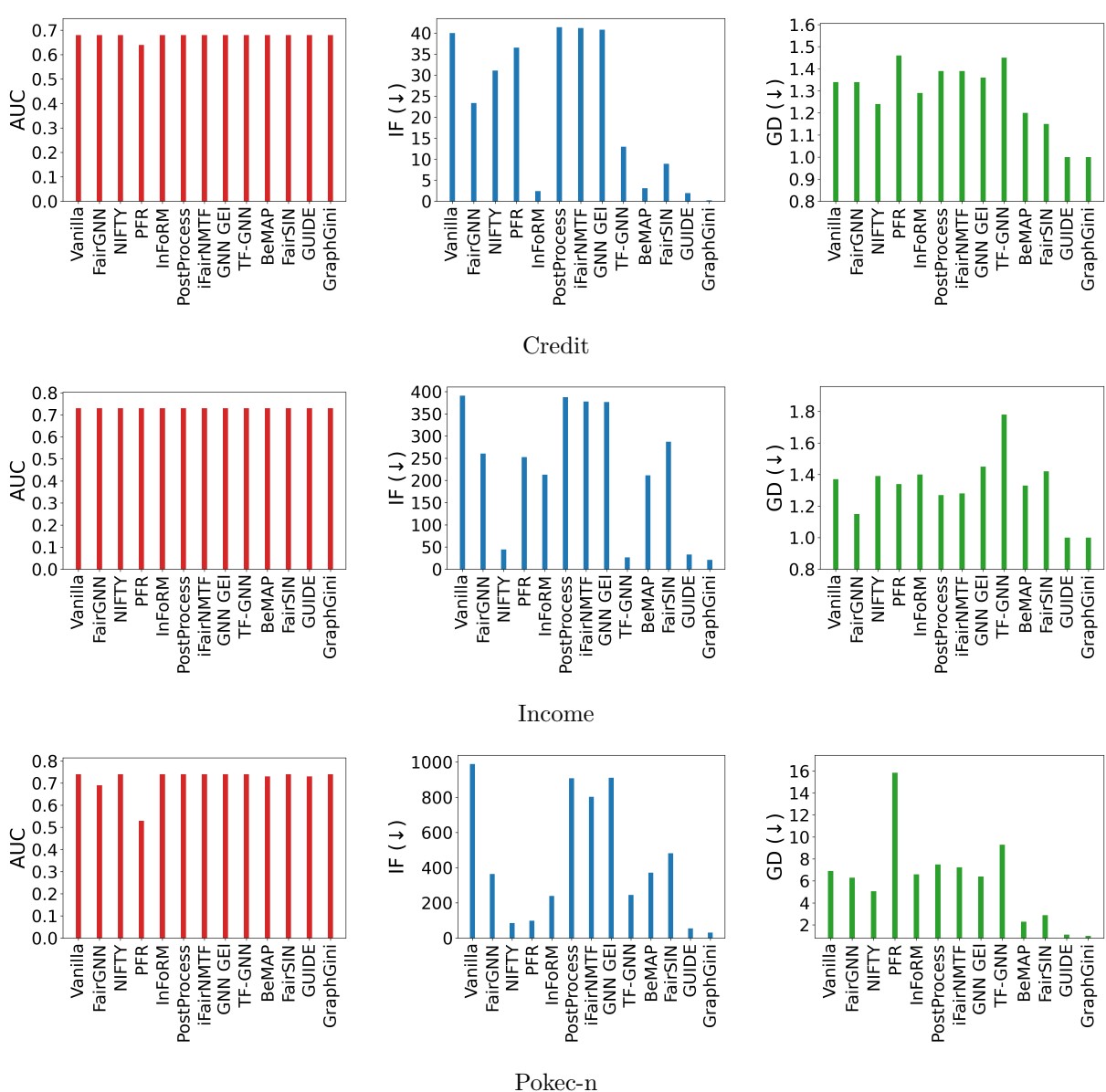

Figure 3: Utility-Fairness Curve comparison of benchmarked algorithms across datasets using the GCN backbone architecture.

Table 9: Running time comparison for GCN on all three datasets. Time is reported in seconds for one iteration.

| Datasets → Model ↓ | Credit | Income | Pokec-n |
|---|---|---|---|
| Vanila | 0.014 | 0.046 | 0.082 |
| GUIDE | 0.041 | 0.051 | 0.094 |
| GraphGini | 0.042 | 0.050 | 0.096 |

incorporate to ensure individual and group fairness. Nonetheless, the running times remain small enough for practical workloads.

## 5 Conclusion

In this work, we have shown how to combine the two key requirements of group fairness and individual fairness in a single GNN architecture GRAPHGINI. The GRAPHGINI achieves individual fairness by employing learnable attention scores that facilitate the aggregation of more information from similar nodes. Our major contribution is that we have used the well-accepted Gini coefficient to define fairness, overcoming the difficulty posed by its non-dfferentiability. This particular way of using the Gini coefficient may be of interest to future research, as it provides a bridge between economic inequality measures and machine learning. Our approach to group fairness incorporates the concept of Nash Social Welfare. We also demonstrate that the gradient-normalization-based technique (GradNorm) provides an effective way to balance utility, individual fairness, and group fairness objectives in graph learning, mitigating the need for extensive manual tuning of objective weights. Empirical findings show GRAPHGINI significantly reduces individual unfairness while maintaining group disparity and utility performance after beating all state-of-the-art existing methods.

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

## A  Notations

To ensure clarity and consistency in the mathematical formulations used throughout this work, Table J summarizes the key notations employed.

Table J: Notations.

| Notation | Description |
|---|---|
| $G$ | input graph |
| $\mathcal{V}$ | set of nodes in graph |
| $\mathcal{E}$ | set of edges in graph |
| $n$ | number of nodes in a graph |
| $d$ | features dimension |
| $\mathcal{V}_h$ | $h^{th}$ group in graph |
| $\mathbf{A} \in \{0,1\}^{n \times n}$ | adjacency matrix of graph $G$ |
| $\mathbf{X} \in \mathbb{R}^{n \times d}$ | node feature matrix of graph $G$ |
| $\mathbf{Z} \in \mathbb{R}^{n \times c}$ | output learning matrix of graph $G$ with $c$ number of features |
| $\mathbf{S} \in \mathbb{R}^{n \times n}$ | pairwise similarity matrix of graph $G$ |
| $\mathbf{L} \in \mathbb{R}^{n \times n}$ | Laplacian similarity matrix |
| $Gini(\mathcal{V})$ | Gini of node set $\mathcal{V}$ |

## B  Lorenz curve

It is the plot of the proportion of the total income of the population (on the y-axis) cumulatively earned by the bottom x (on the x-axis) of the population. The line at 45 degrees thus represents perfect equality of incomes. The further away the Lorenz curve is from the line of perfect equality, the greater the inequality. The Gini coefficient is the area ratio between the line of equality and the Lorenz curve over the total area under the line of quality.

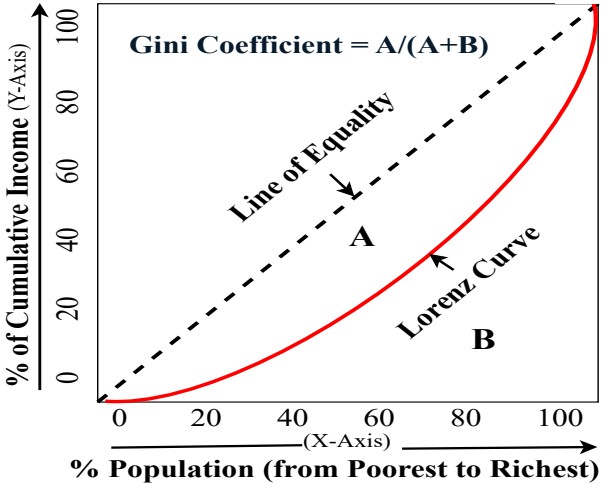

Figure D:  Lorenz curve

## C   Training Procedure of GraphGini

Algorithm 1 outlines the training process of the proposed GRAPHGINI model. Given an input graph comprising nodes, edges, and node features, along with a pairwise similarity matrix, the model aims to learn node embeddings that are both accurate and fair. The optimization involves three loss components—utility, individual fairness, and group fairness—combined using adaptive weights. The training proceeds iteratively through attention-based message passing over $K$ layers, with GradNorm used to dynamically adjust the influence of each loss term. The model parameters, attention coefficients, and loss weights are updated until convergence.

---
**Algorithm 1 GraphGini**

---
**Input**: $G = (\mathcal{V}, \mathcal{E}, \mathbf{X})$, $\mathbf{S}$
**Output**: Fair Embeddings $\mathbf{Z}$

1:  $\mathbf{Z} \leftarrow$ Initialize GNN embeddings of $\mathcal{V}$ using utility loss $\mathcal{L}_1$
2:  Initialize $\beta_1 \leftarrow 1$, $\beta_2 \leftarrow 1$, $\beta_3 \leftarrow 1$, $t \leftarrow 0$
3:  Compute $\mathcal{L}_1$, $\mathcal{L}_2$, and $\mathcal{L} \leftarrow \beta_1 \mathcal{L}_1 + \beta_2 \mathcal{L}_2 + \beta_3 \mathcal{L}_3$
4:  **while** not converged in epoch $t$ **do**
5:     **for** layer $\ell = 1$ to $K$ **do**
6:        **for** each node $v_i \in \mathcal{V}$ **do**
7:           $\mathbf{h}_i^{\ell+1} = \sigma\left(\sum_{j \in \mathcal{N}_i} \alpha_{i,j} \mathbf{W} \mathbf{h}_j^{\ell}\right)$                     `// message passing`
8:        **end for**
9:     **end for**
10:    $\mathcal{L} \leftarrow \beta_1 \mathcal{L}_1 + \beta_2 \mathcal{L}_2 + \beta_3 \mathcal{L}_3$
11:    Backpropagate using GradNorm loss corresponding to $\mathcal{L}$ (Eq. 14)
12:    Update $\mathbf{W}$, $\alpha_{i,j}$, $\beta_1$, $\beta_2$, $\beta_3$, and $\mathbf{Z}$
13:    $t \leftarrow t + 1$
14: **end while**
15: **Return $\mathbf{Z}$**

---

## D   Baselines

- **Guide** Song et al. (2022): This method is the state-of-the-art, which minimizes the average Lipschitz constant for individual fairness and proposes a new group disparity measure based on the ratios of individual fairness among the groups.

- **FairGNN** Dai & Wang (2021): This model leverages adversarial learning to ensure that GNNs achieve fair node classifications, adhering to group fairness criteria.

- **NIFTY** Agarwal et al. (2021): Addressing counterfactual fairness along with stability problem, NIFTY perturbs attributes and employs Lipschitz constants to normalize layer weights. Training incorporates contrastive learning techniques, and we adopt this model directly for various GNN backbone architectures.

- **PFR** Lahoti et al. (2020): PFR learns fair node embeddings as a pre-processing step, ensuring individual fairness in downstream tasks. The acquired embeddings serve as inputs for GNN backbones.

- **InFoRM** Kang et al. (2020): This model formulates an individual fairness loss within a graph framework based on the Lipschitz condition. We integrate the proposed individual fairness loss into the training process of GNN backbones.

- **PostProcessLohia et al. (2019):** Lohia et al. (2019) proposes a post-processing based method to enhance individual and group fairness. The method employs a bias detector to assess disparity in outcomes, and when such biases are detected, it changes the model output to a different outcome. This algorithm is topology-agnostic.

- **GEI Speicher et al. (2018)**: GEI considers diversity of outcomes within a group, determined by sensitive attributes, as a measure of inequality. This implies the assumption that outcomes are independent of non-sensitive attributes within the group. In contrast, Gini allows weighting outcomes proportional to an input similarity measure (Eq. 2), leading to a more nuanced calculation of inequality based on similarity in non-sensitive attributes. To quantify this effect, we use GEI as the regularizer instead of Gini.

- **iFairNMTF Ghodsi et al. (2024)**: iFairNMTF is a fair clustering model that uses individual Fairness Nonnegative Matrix Tri-Factorization technique with contrastive fairness regularization to get balanced and cohesive clusters. We adapt iFairNMTF in our setting by plugging their fairness regularizer term with our GNN loss.

- **TF-GNN Song et al. (2023)**: TF-GNN presents an individual fair Graph Neural Networks (GNNs) tailored for the analysis of temporal financial transaction network data. In our specific context, we integrate TF-GNN into our framework by incorporating their fairness regularizer term into our GNN loss function.

- **BeMap Lin et al. (2024)**: BeMap is a group fair Graph Neural Networks (GNNs) method based on fair message passing algorithm on the basis of 1-hop neighbours from different sensitive groups. We used this method in our setting of different groups.

- **FairSIN Yang et al. (2024)**: FairSIN is also a group fairness enforcing framework for GNNs which proposes a neutralization-centered procedure, and using supplementary Fairness-facilitating Features (F3). These features are integrated into node representations prior to message passing. We also adopt this method to compare our method

## E   Tightness of upper bound in Eq. 8.

*Proof of Proposition 2.* Consider any $\mathbf{Z}$ such that each entry of $\mathbf{Z}$ is identical. In this case both $\sum_{j=1}^{|\mathcal{E}|} \mathbf{S}[i,j]\|\mathbf{z}_i - \mathbf{z}_j\|_1$ and $\sum_{j=1}^{|\mathcal{E}|} \mathbf{S}[i,j]\|\mathbf{z}_i - \mathbf{z}_j\|_2$ are 0 □

Let $\mathbf{z} = (z_1, \ldots, z_c) \in \mathbb{R}^c$ be a $c$-dimensional vector. We consider the relationship between its $\ell_1$ and $\ell_2$ norms. When $\mathbf{z}$ has a single non-zero entry (e.g., $(1, 0, \ldots, 0)$), the bound is loose: $\|\mathbf{z}\|_1 = \|\mathbf{z}\|_2 = 1$, while $\sqrt{c}\|\mathbf{z}\|_2 = \sqrt{c}$. When all entries of $\mathbf{z}$ are equal (e.g., $(1, 1, \ldots, 1)$), the bound is tight: $\|\mathbf{z}\|_1 = c$, $\|\mathbf{z}\|_2 = \sqrt{c}$, and $\sqrt{c}\|\mathbf{z}\|_2 = c$.

Since minimizing $\widehat{Gini}(\mathcal{V})$ minimizes the $\ell_2$-norm, we have: $\min_{\mathbf{z}}(\widehat{Gini}(\mathcal{V})) \implies \min_{\mathbf{z}} \|\mathbf{z}\|_2$ While this does not directly minimize $\|\mathbf{z}\|_1$, we can establish: $\min_{\mathbf{z}} \|\mathbf{z}\|_2 \implies \min_{\mathbf{z}} \sqrt{c}\|\mathbf{z}\|_2 \geq \min_{\mathbf{z}} \|\mathbf{z}\|_1$ This inequality shows that minimizing $\|\mathbf{z}\|_2$ provides an upper bound on the minimum of $\|\mathbf{z}\|_1$. As $\|\mathbf{z}\|_2 \to 0$ during optimization, both $\|\mathbf{z}\|_1$ and $\sqrt{c}\|\mathbf{z}\|_2$ approach 0.

**Lemma 1.** *The upper bound of the Gini coefficient, defined as*

$$\widehat{Gini}(\mathcal{V}) = \mathrm{Tr}(\mathbf{Z}^T \mathbf{L} \mathbf{Z}),$$

*is a convex function in $\mathbf{Z}$.*

*Proof.* To prove convexity, we show that the Hessian of $\widehat{Gini}(\mathcal{V})$ is positive semidefinite. The function is given by:

$$f(\mathbf{Z}) = \mathrm{Tr}(\mathbf{Z}^T \mathbf{L} \mathbf{Z}) = \sum_{i=1}^{n} \sum_{j=1}^{n} \mathbf{S}[i,j]\|\mathbf{z}_i - \mathbf{z}_j\|_2^2.$$

This is a quadratic form defined by the Laplacian $\mathbf{L} = \mathbf{D} - \mathbf{S}$, where $\mathbf{D}$ is the diagonal matrix.

**Gradient Computation:** Taking the gradient with respect to $\mathbf{Z}$:

$$\nabla f(\mathbf{Z}) = 2\mathbf{L}\mathbf{Z}.$$

**Hessian Analysis:** The Hessian of $f(\mathbf{Z})$ is:

$$H = 2\mathbf{L} \otimes \mathbf{I}_d,$$

where $\otimes$ denotes the Kronecker product, and $\mathbf{I}_d$ is the $d \times d$ identity matrix.

**Positive Semidefiniteness of Hessian:** The Laplacian $\mathbf{L}$ is known to be positive semidefinite, meaning for any vector $\mathbf{v} \in \mathbb{R}^n$,

$$\mathbf{v}^T \mathbf{L} \mathbf{v} \geq 0.$$

Since the Kronecker product of a positive semidefinite matrix with $\mathbf{I}_d$ preserves positive semidefiniteness, we conclude that:

$$H = 2\mathbf{L} \otimes \mathbf{I}_d \succeq 0.$$

This proves that $f(\mathbf{Z})$ is a convex function in $\mathbf{Z}$. $\qquad\square$

### E.1 The need for differentiability

For deep learning methods, if the loss function is differentiable, then its gradient can be obtained using inbuilt packages such as PyTorch to update the model's weights. In practice, these packages can handle the non-differentiability of activation functions such as ReLU by using hand-coded values at non-differentiable points. However, such values for generic loss functions are not available a priori; therefore, differentiability of the loss function is essential. To elaborate, consider the Gini index. If $\mathbf{x} = (x_1, \ldots, x_n)$ and $f(\mathbf{x}) = \sum_{i=1}^n \sum_{j=1}^n |x_i - x_j|$, for a fixed $j \in [n]$, let us define

$$m_j(\mathbf{x}) = |\{i \in [n] : i \neq j, x_i < x_j\}|.$$

In the case where $x_i \neq x_j$ for all $i \neq j$, clearly

$$\frac{\partial f(\mathbf{x})}{\partial x_j} = n - 1 - 2m_j(\mathbf{x}).$$

Let $k$ be the index such that $x_k < x_j$ and there is no other coordinate value between them. Suppose we decrease $x_j$ until it reaches $x_k$, then we reach a point of discontinuity, which we will call $\mathbf{y}$. We note that the left partial derivative with respect to $x_j$ is $n - 3 - 2m_j(\mathbf{x})$, which is 2 less than its right partial derivative. Unlike the case of ReLU, there is no straightforward way to decide which of these two values should be assumed at the point of non-differentiability. Furthermore, unlike ReLU, these two values cannot be determined a priori since they depend on $\mathbf{x}$.

## F Pareto Optimality Proof

*Proof of Proposition 3.* Let the Gini upper bounds minimizing Eq. 9 be $\widehat{Gini}(\mathcal{V}_g) = \epsilon_g$ and $\widehat{Gini}(\mathcal{V}_h) = \epsilon_h$. The solution is Pareto optimal if we cannot reduce $\epsilon_h$ without increasing $\epsilon_g$ and vice-versa. We establish Pareto optimality through proof by contradiction. Suppose that there exists an assignment of node embeddings for which we get $\widehat{Gini}(\mathcal{V}_g) = \epsilon_g^*$, $\widehat{Gini}(\mathcal{V}_h) = \epsilon_h$ and $\epsilon_g^* > \epsilon_g$. Since Eq. 9 is minimized at $\epsilon_g$ and $\epsilon_h$, this means that $-\left(\frac{\epsilon_g}{\epsilon_h} - 1\right)\left(\frac{\epsilon_h}{\epsilon_g} - 1\right) < -\left(\frac{\epsilon_g^*}{\epsilon_h} - 1\right)\left(\frac{\epsilon_g}{\epsilon_h} - 1\right)$, which is a contradiction. $\qquad\square$

## G Convergence and Time Complexity

**Convergence.** As our framework considers only standard GNNs, the convergence analysis of gradient descent on GNNs has been conducted in Awasthi et al. (2021) under the assumption of the smoothness of a bounded loss function. Our overall loss is smooth, as the fairness regularizers for individual and group fairness are designed to ensure differentiability. Additionally, each term is bounded (loss $\mathcal{L}_1$: CE loss, $\mathcal{L}_2 \leq tr(\mathbf{X}^T L \mathbf{X})$, and $\mathcal{L}_3 \leq g$ [finite value of group disparity]); hence, the overall loss is smooth and bounded. Therefore, the analysis in Awasthi et al. (2021) directly applies to our framework.

**Runtime.** Table 9 presents the runtime for GraphGini and the baselines. Empirical analysis shows that the inference time (i.e., the time taken for a forward pass) is comparable between GraphGini, GradNorm,

and GUIDE across all three datasets tested. Nonetheless, it's important to note that the absolute inference times for all methods are very small, suggesting that the computational overhead should not be a significant concern for the practical deployment of these fair GNN approaches.

Table K: Baseline hyper-parameters. − indicates the parameter not used to train the model.

| Datasets → | Credit | | Income\| | | Pokec-n | |
|---|---|---|---|---|---|---|
| Model ↓ | $\beta_2$ | $\beta_3$ | $\beta_2$ | $\beta_3$ | $\beta_2$ | $\beta_2$ |
| **FairGNN** | 4 | 1000 | 4 | 10 | 4 | 100 |
| **NIFTY** | - | - | - | - | - | - |
| **PFR** | - | - | - | - | - | - |
| **PostProcess** | - | - | - | - | - | - |
| **iFairNMTF** | 1e-7 | - | 1e-7 | - | 1e-7 | - |
| **GNN GEI** | 1 | - | 1 | - | 1 | - |
| **TF-GNN** | 1e-6 | - | 1e-7 | - | 1e-7 | - |
| **InFoRM** | 5e-6 | - | 1e-7 | - | 1e-7 | - |
| **GUIDE** | 5e-6 | 1 | 1e-7 | 0.25 | 2.5e-7 | 0.05 |

## H   Implementation details for Reproducibility

Each experiment is conducted five times, and the reported results consist of averages accompanied by standard deviations. Our experiments are performed on a machine with Intel(R) Core(TM) CPU @ 2.30GHz with 16GB RAM, RTX A4000 GPU having 16GB memory on Microsoft Windows 11 HSL.

For all three datasets, we employ a random node shuffling approach and designate 25% of the labeled nodes for validation and an additional 25% for testing purposes. The training set sizes are set at 6,000 labeled nodes (25%) for the Credit dataset, 3,000 labeled nodes (20%) for the Income dataset, and 4,398 labeled nodes (6%) for Pokec-n. For the Pokec-n dataset, friendship linkages serve as edges, while for the remaining datasets, edges are not predefined, necessitating their construction based on feature similarity. More precisely, we establish a connection for any given pair of nodes if the Euclidean distances between their features surpass a predetermined threshold. The fine-tuned hyper-parameters used to train baselines are given in Table K. $\beta_1$ is 1 for all baselines. For GRAPHGINI, the backbone GNN architecture parameters are the same for all datasets, i.e. one hidden layer with hidden dimensions 16.

## I   Number of clusters in datasets

To evaluate the Gini, we divide the test dataset into a number of clusters based on the elbow graph (Figure. E). Specifically, we employ K-means clustering to group samples based on their features, aiming to capture structural or distributional disparities that may exist in the data. The number of clusters for each dataset is determined using the elbow method, which evaluates the within-cluster sum of squares (WCSS) as a function of cluster count. As shown in Figure E, the elbow point—where the marginal gain in reducing WCSS begins to diminish—indicates the optimal number of clusters for the Credit, Income, and Pokec-n datasets.

Table L presents a detailed comparison of the proposed GRAPHGINI model against baseline methods across three benchmark datasets. The results include performance metrics such as AUC and F1-score, alongside fairness indicators—individual fairness (IF), group disparity (GD), and Gini coefficients computed for multiple subpopulations (clusters). The term Vanilla refers to a baseline model trained without any fairness enhancement. Across all architectures (GCN, GIN, JK), GRAPHGINI consistently achieves comparable or superior predictive performance while significantly reducing fairness violations, as reflected in both the IF and Gini values.

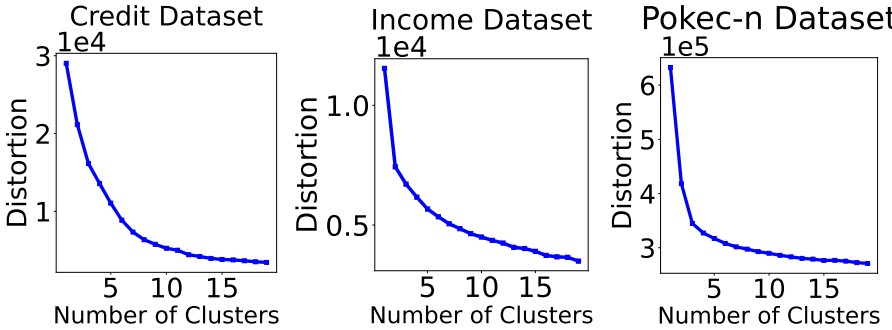

Figure E: The Elbow plots show the optimal number of clusters in each dataset on K-means clustering.

## J   Impact of Gradient Normalization

In Fig. F, we plot the trajectory of the weights against training epochs. In the credit dataset, initially, utility loss is given higher weightage than group fairness loss, but after certain iterations, group fairness loss weightage overcomes utility loss weightage. Meanwhile, the utility loss is always given the higher weightage in the other two datasets. These behaviors indicate the sensitivity of $\beta_i$. With the increase in the number of iterations/epochs, initially, $\beta_i$s' are changing at different rates, but after a certain iteration, the values of $\beta_i$s' get stabilized, indicating similar training rates across all three losses.

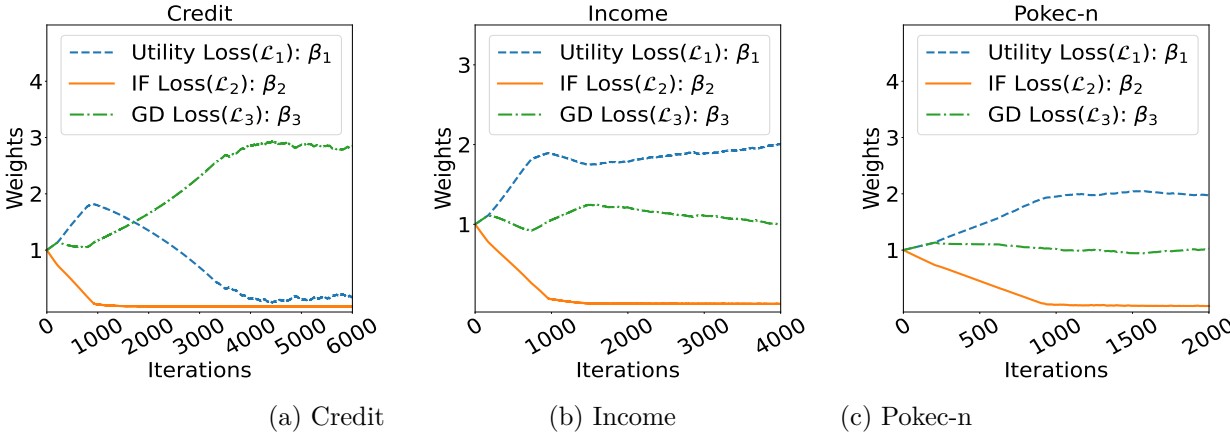

(a) Credit                    (b) Income                    (c) Pokec-n

Figure F: Adaptive loss weights learned via GradNorm during training for the GRAPHGINI model with a GCN backbone across three datasets.

## K   Multidimensional Gini and Comparison of its Differentiable Upper Bound with Smooth Pooling Operators

Let $\Delta(Z) \in \mathbb{R}_{\geq 0}^M$ collect pairwise embedding disparities $\Delta_{ij} := \|z_i - z_j\|$ for $M = \binom{n}{2}$ unordered pairs. Define the (unnormalized) *Gini fairness* functional

$$\mathcal{G}ini(Z) \;=\; \frac{1}{M} \sum_{(i,j)} \Delta_{ij}, \tag{15}$$

i.e., the mean absolute pairwise disparity. By contrast, conventional pooling surrogates optimize

$$\frac{1}{M} \sum_{i,j} \mathbb{I}[\Delta_{ij} \geq \varepsilon] \;. \tag{16}$$

Table L: Gini coefficient for different clusters on Credit, Income, and Pokec-n datasets. The model indicates the algorithm, and Vanilla represents that no fairness mechanism has been used. Best performances are in bold. Individual (un)fairness numbers are reported in thousands.

| Model → | Vanilla | Guide | GraphGini | Vanilla | Guide | GraphGini | Vanilla | Guide | GraphGini |
|---|---|---|---|---|---|---|---|---|---|
| | | GCN | | | GIN | | | JK | |
| **Credit** | | | | | | | | | |
| **AUC(↑)** | 0.68 | 0.67 | **0.69** | **0.71** | 0.69 | 0.69 | 0.69 | 0.68 | 0.68 |
| **F1-score(↑)** | 0.71 | 0.72 | **0.73** | 0.72 | 0.73 | **0.74** | 0.70 | 0.71 | **0.73** |
| **IF(↓)** | 17.38 | 1.09 | **1.01** | 27.47 | 1.97 | **1.78** | 15.55 | 1.48 | **1.42** |
| **GD(↓)** | 1.35 | **1.00** | **1.00** | 1.87 | **1.00** | **1.00** | 1.24 | **1.00** | **1.00** |
| **Gini Cl 1 (↓)** | 0.08 | **0.04** | 0.04 | 0.11 | 0.08 | **0.05** | 0.11 | 0.04 | **0.03** |
| **Gini Cl 2 (↓)** | 0.10 | 0.08 | **0.08** | 0.13 | 0.04 | **0.04** | 0.14 | **0.04** | 0.05 |
| **Gini Cl 3 (↓)** | 0.27 | **0.26** | 0.26 | 0.29 | 0.06 | **0.06** | 0.25 | **0.07** | 0.08 |
| **Gini Cl 4 (↓)** | 0.20 | 0.19 | **0.19** | 0.17 | 0.13 | **0.10** | 0.19 | 0.08 | **0.08** |
| **Gini Cl 5 (↓)** | 0.09 | 0.07 | **0.07** | 0.11 | **0.04** | 0.05 | 0.13 | 0.04 | **0.04** |
| **Gini Cl 6 (↓)** | 0.08 | 0.05 | **0.05** | 0.11 | **0.05** | 0.06 | 0.13 | 0.03 | **0.03** |
| **Income** | | | | | | | | | |
| **AUC(↑)** | 0.77 | 0.74 | **0.78** | **0.81** | 0.80 | 0.80 | **0.80** | 0.74 | 0.75 |
| **F1-score(↑)** | 0.78 | **0.79** | 0.78 | 0.80 | **0.81** | **0.81** | 0.79 | **0.80** | **0.80** |
| **IF(↓)** | 111.03 | 9.75 | **6.85** | 421.81 | 23.46 | **22.31** | 439.38 | 27.14 | **24.39** |
| **GD(↓)** | 1.23 | **1.00** | **1.00** | 1.16 | **1.00** | **1.00** | 1.29 | **1.00** | **1.00** |
| **Gini Cl 1 (↓)** | 0.34 | 0.19 | **0.17** | 0.43 | **0.16** | 0.17 | 0.40 | **0.11** | 0.11 |
| **Gini Cl 2 (↓)** | 0.28 | 0.18 | **0.18** | 0.38 | 0.02 | 0.17 | 0.34 | **0.10** | 0.15 |
| **Gini Cl 3 (↓)** | 0.32 | 0.15 | **0.12** | 0.56 | 0.15 | **0.15** | 0.47 | **0.12** | 0.16 |
| **Gini Cl 4 (↓)** | 0.21 | 0.08 | **0.07** | 0.21 | 0.15 | **0.15** | 0.24 | 0.12 | **0.11** |
| **Gini Cl 5 (↓)** | 0.37 | **0.13** | 0.13 | 0.60 | 0.25 | **0.24** | 0.49 | 0.39 | **0.32** |
| **Gini Cl 6 (↓)** | 0.22 | **0.12** | 0.13 | 0.24 | **0.16** | 0.16 | 0.25 | 0.16 | **0.13** |
| **Pokec-n** | | | | | | | | | |
| **AUC(↑)** | **0.77** | 0.74 | 0.74 | **0.76** | 0.74 | 0.74 | **0.79** | 0.78 | 0.78 |
| **F1-score(↑)** | 0.75 | 0.77 | **0.79** | 0.76 | 0.77 | **0.79** | 0.76 | **0.78** | **0.78** |
| **IF(↓)** | 859.67 | 46.60 | **26.16** | 1589.50 | 96.90 | **33.70** | 1450.98 | 59.42 | **44.51** |
| **GD(↓)** | 2.55 | **1.00** | **1.00** | 3.50 | **1.00** | **1.00** | 3.07 | **1.00** | **1.00** |
| **Gini Cl 1** | 0.27 | **0.10** | 0.10 | 0.22 | 0.11 | **0.10** | 0.38 | 0.08 | **0.07** |
| **Gini Cl 2** | 0.17 | 0.09 | **0.08** | 0.13 | **0.13** | 0.14 | 0.34 | 0.09 | **0.08** |
| **Gini Cl 3** | 0.06 | 0.03 | **0.02** | 0.09 | **0.06** | 0.06 | 0.07 | 0.05 | **0.04** |
| **Gini Cl 4** | 0.27 | **0.11** | 0.13 | 0.15 | 0.10 | **0.09** | 0.35 | 0.07 | **0.06** |
| **Gini Cl 5** | 0.16 | 0.07 | **0.06** | 0.11 | **0.11** | 0.11 | 0.09 | 0.07 | **0.07** |
| **Gini Cl 6** | 0.27 | **0.10** | 0.11 | 0.21 | 0.10 | **0.09** | 0.08 | 0.05 | **0.05** |
| **Gini Cl 7** | 0.28 | 0.12 | **0.10** | 0.24 | 0.10 | **0.10** | 0.33 | **0.10** | 0.11 |
| **Gini Cl 8** | 0.27 | 0.16 | **0.12** | 0.29 | 0.12 | **0.09** | 0.33 | 0.10 | **0.08** |

for some $\varepsilon$. However, for any $\varepsilon > 0$, using Markov's inequality on the nonnegative random variable $\Delta_{ij}$ sampled uniformly from pairs yields $\Pr[\Delta_{ij} \geq \varepsilon] \leq \mathbb{E}[\Delta_{ij}]/\varepsilon = \mathcal{G}(Z)/\varepsilon$. Hence, we have,

$$\frac{1}{M} \sum_{i,j} \mathbb{I}[\Delta_{ij} \geq \varepsilon] \ \leq \ \frac{\mathcal{G}ini(Z)}{\varepsilon} \ . \tag{17}$$

Hence, minimizing $\mathcal{G}ini(Z)$ directly controls the *fraction* of unfair pairs at any scale $\varepsilon$. Pooling substitutes (max, smooth-max, top-$k$, $p$-norm) do not provide such calibrated distributional guarantees, they may keep the maximum small while still allowing a large fraction of moderate violations. Further, let $L$ be the graph Laplacian built from the same pairwise weights used in fairness evaluation. Then

$$\mathcal{G}ini(Z) \ \leq \ C \cdot \mathrm{Tr}(Z^\top L Z) \tag{18}$$

Table M: Performance comparison of benchmarked algorithms across datasets using the JK backbone architecture. "Vanilla" indicates no debiasing. Arrows: $\uparrow$ = higher is better, $\downarrow$ = lower is better. Best performance per column is in **bold**. Individual fairness (IF) is reported in thousands.

| Model | AUC ($\uparrow$) | IF ($\downarrow$) | GD ($\downarrow$) | IF-Gini ($\downarrow$) | GD-Gini ($\downarrow$) |
|---|---|---|---|---|---|
| **Credit** | | | | | |
| Vanilla | 0.64±0.10 | 31.50±13.25 | 1.35±0.05 | 0.28 ± 0.03 | 1.71 ± 0.01 |
| FairGNN | 0.66±0.05 | 2.65±1.85 | 1.54±0.28 | 0.19 ± 0.02 | 1.82 ± 0.02 |
| NIFTY | 0.69±0.02 | 30.12±2.25 | 1.26±0.02 | 0.27 ± 0.01 | 1.73 ± 0.00 |
| PFR | 0.67±0.02 | 36.24±19.10 | 1.40±0.02 | 0.27 ± 0.01 | 1.65 ± 0.01 |
| InFoRM | 0.67±0.04 | 5.70±5.2 | 1.46±0.14 | 0.16 ± 0.02 | 1.44 ± 0.00 |
| PostProcess | **0.70±0.00** | 43.57±7.80 | 1.44±0.01 | 0.28 ± 0.01 | 1.69 ± 0.01 |
| iFairNMTF | **0.71±0.01** | 107.72±9.04 | 1.54±0.23 | 0.28 ± 0.02 | 1.66 ± 0.02 |
| GNN GEI | 0.68±0.00 | 43.16 ± 4.15 | 1.41±0.01 | 0.26 ± 0.00 | 1.53 ± 0.02 |
| TF-GNN | **0.70±0.00** | 13.20±0.03 | 1.43±0.01 | 0.20 ± 0.01 | 1.49 ± 0.02 |
| BeMAP | 0.68±0.00 | 11.36±2.07 | 1.45±0.0 | 0.22 ± 0.03 | 1.63 ± 0.01 |
| FairSIN | 0.68±0.00 | 18.39±8.80 | 1.56±0.0 | 0.21 ± 0.02 | 1.56 ± 0.02 |
| GUIDE | 0.68±0.00 | 2.35±0.10 | **1.00±0.00** | 0.12 ± 0.02 | 1.30 ± 0.01 |
| GRAPHGINI | 0.68±0.00 | **1.88±0.02** | **1.00±0.00** | **0.10 ± 0.01** | **1.08 ± 0.01** |
| **Income** | | | | | |
| Vanilla | **0.80±0.01** | 490.70±165.33 | 1.18±0.15 | 0.38 ± 0.02 | 1.80 ± 0.00 |
| FairGNN | 0.77±0.01 | 230.12±40.88 | 1.32±0.14 | 0.35 ± 0.01 | 1.81 ± 0.01 |
| NIFTY | 0.73±0.01 | 47.40±12.20 | 1.40±0.05 | 0.20 ± 0.02 | 1.83 ± 0.01 |
| PFR | 0.73±0.12 | 330.40±150.25 | 1.13±0.21 | 0.36 ± 0.02 | 1.72 ± 0.01 |
| InFoRM | 0.79±0.00 | 195.61±11.78 | 1.36±0.14 | 0.33 ± 0.01 | 1.81 ± 0.02 |
| PostProcess | 0.79±0.00 | 520.23±20 | 1.27±0.01 | 0.40 ± 0.02 | 1.54 ± 0.01 |
| iFairNMTF | 0.78±0.01 | 604.89±24.32 | 1.43±0.26 | 0.44± 0.01 | 1.76 ± 0.03 |
| GNN GEI | 0.79±0.00 | 497.29±19.34 | 1.37±0.12 | 0.37 ± 0.02 | 1.84 ± 0.02 |
| TF-GNN | 0.78±0.01 | 205.08±03.25 | 1.48±0.01 | 0.34 ± 0.01 | 1.90 ± 0.01 |
| BeMAP | 0.75±0.00 | 297.25±41.65 | 1.39±0.0 | 0.35 ± 0.01 | 1.83 ± 0.02 |
| FairSIN | 0.75±0.00 | 282.45±49.55 | 1.46±0.0 | 0.33 ± 0.01 | 1.91 ± 0.01 |
| GUIDE | 0.74±0.01 | 42.50±22.10 | **1.00±0.00** | 0.22 ± 0.02 | 1.13 ± 0.00 |
| GRAPHGINI | 0.75±0.00 | **29.47± 4.01** | **1.00±0.00** | **0.18 ± 0.01** | **1.01 ± 0.01** |
| **Pokec-n** | | | | | |
| Vanilla | **0.79±0.01** | 1639.30±95.74 | 8.48±0.51 | 0.42 ± 0.01 | 1.77 ± 0.01 |
| FairGNN | 0.70±0.00 | 807.79±281.26 | 11.68±2.89 | 0.35 ± 0.01 | 1.79 ± 0.02 |
| NIFTY | 0.73±0.01 | 477.31±165.68 | 8.20±1.33 | 0.31 ± 0.01 | 1.76 ± 0.01 |
| PFR | 0.68±0.00 | 729.77±74.62 | 15.66±5.47 | 0.35 ± 0.02 | 1.80 ± 0.03 |
| InFoRM | 0.78±0.01 | 315.27±25.21 | 6.80±0.54 | 0.30 ± 0.01 | 1.74 ± 0.01 |
| PostProcess | 0.78±0.00 | 1721.42±83.91 | 10.22±0.43 | 0.42 ± 0.03 | 1.80 ± 0.01 |
| iFairNMTF | 0.77±0.00 | 1602.52±92.73 | 9.37±0.10 | 0.41 ± 0.01 | 1.79 ± 0.03 |
| GNN GEI | 0.78±0.00 | 1788.65±56.39 | 9.21±0.55 | 0.43 ± 0.01 | 1.80 ± 0.01 |
| TF-GNN | 0.76±0.00 | 418.31±54.26 | 10.20±1.45 | 0.30 ± 0.02 | 1.81 ± 0.02 |
| BeMAP | 0.76±0.00 | 456.95±125.25 | 5.56±0.45 | 0.30 ± 0.01 | 1.53 ± 0.01 |
| FairSIN | 0.76±0.00 | 282.45±49.55 | 1.46±0.0 | 0.26 ± 0.01 | 1.27 ± 0.01 |
| GUIDE | 0.75±0.02 | 83.09±18.70 | 1.13±0.02 | 0.25 ± 0.01 | 1.25 ± 0.01 |
| GRAPHGINI | 0.78±0.10 | **43.87±2.36** | **1.00± 0.00** | **0.23 ± 0.01** | **1.18 ± 0.01** |

for an explicit constant $C > 0$ depending only on the weight normalization and feature bounds. By Jensen's inequality and the Cauchy–Schwarz bound $\|u\| \leq \sqrt{\|u\|^2}$ relate the mean absolute disparity to the mean

squared disparity; the latter equals a Laplacian quadratic form. Therefore $\mathcal{G}ini(Z)$ admits a convex, smooth upper bound in $\text{Tr}(Z^\top L Z)$.

To further validate our approach empirically, we compare GRAPHGINI with smooth pooling-operator surrogates. Specifically, we employ two commonly used surrogates, softmax and top-$k$ as individual-fairness regularizers in conjunction with the proposed Gini-based individual fairness. We conduct an ablation study on all three datasets using identical train–validation–test splits and a GCN backbone. The results are presented in Tables N, O, and P. These results clearly indicate that modifying pooling surrogates alone is insufficient to achieve fair graph embeddings. It is also important to emphasize that group fairness plays a crucial role in graph learning alongside individual fairness, as demonstrated in Table 6 of the manuscript.

Table N: Performance comparison of pooling surrogates and GRAPHGINI on Credit dataset. Arrows: ↑ higher better, ↓ lower better.

| Method | AUC (↑) | IF (↓) | GD (↓) | IF-Gini (↓) | GD-Gini (↓) |
|---|---|---|---|---|---|
| Softmax | 0.68±0.01 | 5.12±0.01 | 1.20±0.02 | 0.24±0.01 | 1.11±0.03 |
| Top-$k$ (5%) | 0.68±0.01 | 19.26±0.01 | 1.34±0.01 | 0.25±0.01 | 1.11±0.01 |
| GRAPHGINI | 0.68±0.01 | **0.22**±0.01 | 1.01±0.01 | **0.12**±0.00 | 1.09±0.01 |

Table O: Performance comparison of pooling surrogates and GRAPHGINI on Income dataset. Arrows: ↑ higher better, ↓ lower better.

| Method | AUC (↑) | IF (↓) | GD (↓) | IF-Gini (↓) | GD-Gini (↓) |
|---|---|---|---|---|---|
| Softmax | 0.73±0.01 | 290.1±6.70 | 1.33±0.01 | 0.21±0.01 | 1.19±0.01 |
| Top-$k$ (5%) | 0.73±0.01 | 213.4±8.04 | 1.25±0.01 | 0.22±0.01 | 1.17±0.01 |
| GRAPHGINI | 0.73±0.01 | **21.12**±5.20 | **1.00**±0.01 | **0.17**±0.00 | **1.03**±0.01 |

Table P: Performance comparison of pooling surrogates and GRAPHGINI on Pokec-n dataset. Arrows: ↑ higher better, ↓ lower better.

| Method | AUC (↑) | IF (↓) | GD (↓) | IF-Gini (↓) | GD-Gini (↓) |
|---|---|---|---|---|---|
| Softmax | 0.74±0.01 | 345.81±8.9 | 3.07±0.02 | 0.31±0.01 | 1.15±0.01 |
| Top-$k$(5%) | 0.74±0.01 | 276.57±11.0 | 5.09±0.02 | 0.29±0.01 | 1.18±0.01 |
| GRAPHGINI | 0.74±0.01 | **31.10**±5.19 | **1.00**±0.01 | **0.21**±0.01 | **1.14**±0.01 |

## L   Comparison with Ranking-Based Approach

We also compared against REDRESS Dong et al. (2021), which is a ranking-based approach. As shown in the Table Q, GRAPHGINI consistently outperforms REDRESS by a significant margin, demonstrating the effectiveness of our framework even beyond the Lipschitz setting.

## M   Hyperparameter Sensitivity Analysis

In the GRAPHGINI framework, the two key parameters, $\beta_2$ and $\beta_3$, must be fine-tuned to achieve optimal performance while balancing both fairness objectives. We evaluate their impact on the performance of GRAPHGINI by varying these parameters simultaneously and independently. Figure G illustrates the effect of GRAPHGINI regularizers on accuracy (AUC), individual fairness (IF), and group fairness (GD) on the Credit dataset using the GCN backbone. In Figures (a-c), the x-axis represents $\beta_2$, the y-axis represents $\beta_3$, and the z-axis shows AUC, IF, and GD in Figures (a), (b), and (c), respectively. The trends indicate that increasing both $\beta_2$ and $\beta_3$ results in a decrease in AUC, while individual and group fairness improves. In all cases, $\beta_1$ is fixed at 1, and lower IF and GD values correspond to better individual and group fairness, respectively.

Table Q: Performance comparison of REDRESS and GRAPHGINI across datasets using the GCN backbone architecture. Arrows: ↑ = higher is better, ↓ = lower is better. Individual fairness (IF) is reported in thousands.

| Model | AUC (↑) | IF (↓) | GD (↓) | IF-Gini (↓) | GD-Gini (↓) |
|---|---|---|---|---|---|
| **Credit** | | | | | |
| REDRESS | 0.68±0.01 | 2.33±0.01 | 1.52±0.01 | 0.17±0.01 | 1.77±0.01 |
| GRAPHGINI | 0.68±0.00 | 0.22±0.06 | 1.00±0.00 | 0.12±0.01 | 1.10±0.01 |
| **Income** | | | | | |
| REDRESS | 0.73±0.03 | 187.10±0.02 | 1.39±0.02 | 0.25±0.01 | 1.56±0.02 |
| GRAPHGINI | 0.73±0.09 | 21.12±5.22 | 1.00±0.00 | 0.17±0.01 | 1.02±0.01 |
| **Pokec-n** | | | | | |
| REDRESS | 0.74±0.00 | 350.22±9.10 | 7.10±0.15 | 0.39±0.02 | 1.85±0.02 |
| GRAPHGINI | 0.74±0.00 | 31.10±5.22 | 1.00±0.00 | 0.21±0.01 | 1.14±0.01 |

Figures (d-i) present when one fairness regularizer is varied while the other is fixed at 0. In Figures (d-f), $\beta_3$ is set to 0, showing that increasing $\beta_2$ decreases accuracy but improves individual fairness. However, group fairness may fluctuate since optimization focuses solely on individual fairness. In Figures (g-i), $\beta_2$ is set to 0, demonstrating that increasing $\beta_3$ reduces accuracy but enhances group fairness, with potential variations in individual fairness due to the focus on group fairness optimization.

## N  Sensitivity to Homophily/Heterophily

We evaluate the robustness of GRAPHGINI under varying levels of structural homophily on all three datasets: Credit, Income, and Pokec-n. The degree of homophily is controlled through a degree-preserving edge rewiring. For each value of $\rho \in \{0, 0.2, 0.4, 0.6, 0.8\}$, a fraction $\rho|E|$ of edges is randomly selected, and one endpoint of each edge is replaced with a node having the same class label as the source. Increasing $\rho$ progressively raises the proportion of same-label connections, thereby producing graphs with higher homophily while maintaining the original degree distribution. For every generated graph, we compute both topological and attribute-based similarity matrices $S$ as defined in Sec. 2. We performed all experiments using the GCN backbone (the same trends hold for GIN and JK architectures) and report the results in Tables R, S, and T. It is evident that GRAPHGINI consistently maintains fairness and accuracy under diverse structural regimes. This validates GRAPHGINI 's robustness under varying graph structures.

Table R: Performance of GRAPHGINI compared GUIDE with varying homophily levels ($\rho$) on the Credit dataset using the GCN backbone. Arrows: ↑ higher is better, ↓ lower is better.

| $\rho$ | Method | AUC (↑) | IF (↓) | GD (↓) | IF-Gini (↓) | GD-Gini (↓) |
|---|---|---|---|---|---|---|
| 0.0 | GUIDE | 0.68 | 1.98 | 1.08 | 0.16 | 1.41 |
| | GRAPHGINI | 0.68 | 0.42 | 1.08 | 0.14 | 1.12 |
| 0.2 | GUIDE | 0.68 | 1.96 | 1.08 | 0.16 | 1.40 |
| | GRAPHGINI | 0.68 | 0.31 | 1.06 | 0.13 | 1.11 |
| 0.4 | GUIDE | 0.68 | 1.94 | 1.05 | 0.14 | 1.33 |
| | GRAPHGINI | 0.68 | 0.25 | 1.03 | 0.12 | 1.10 |
| 0.6 | GUIDE | 0.69 | 1.94 | 1.05 | 0.14 | 1.23 |
| | GRAPHGINI | 0.69 | 0.22 | 1.01 | 0.12 | 1.09 |
| 0.8 | GUIDE | 0.69 | 1.94 | 1.01 | 0.14 | 1.20 |
| | GRAPHGINI | 0.69 | 0.21 | 1.00 | 0.11 | 1.09 |

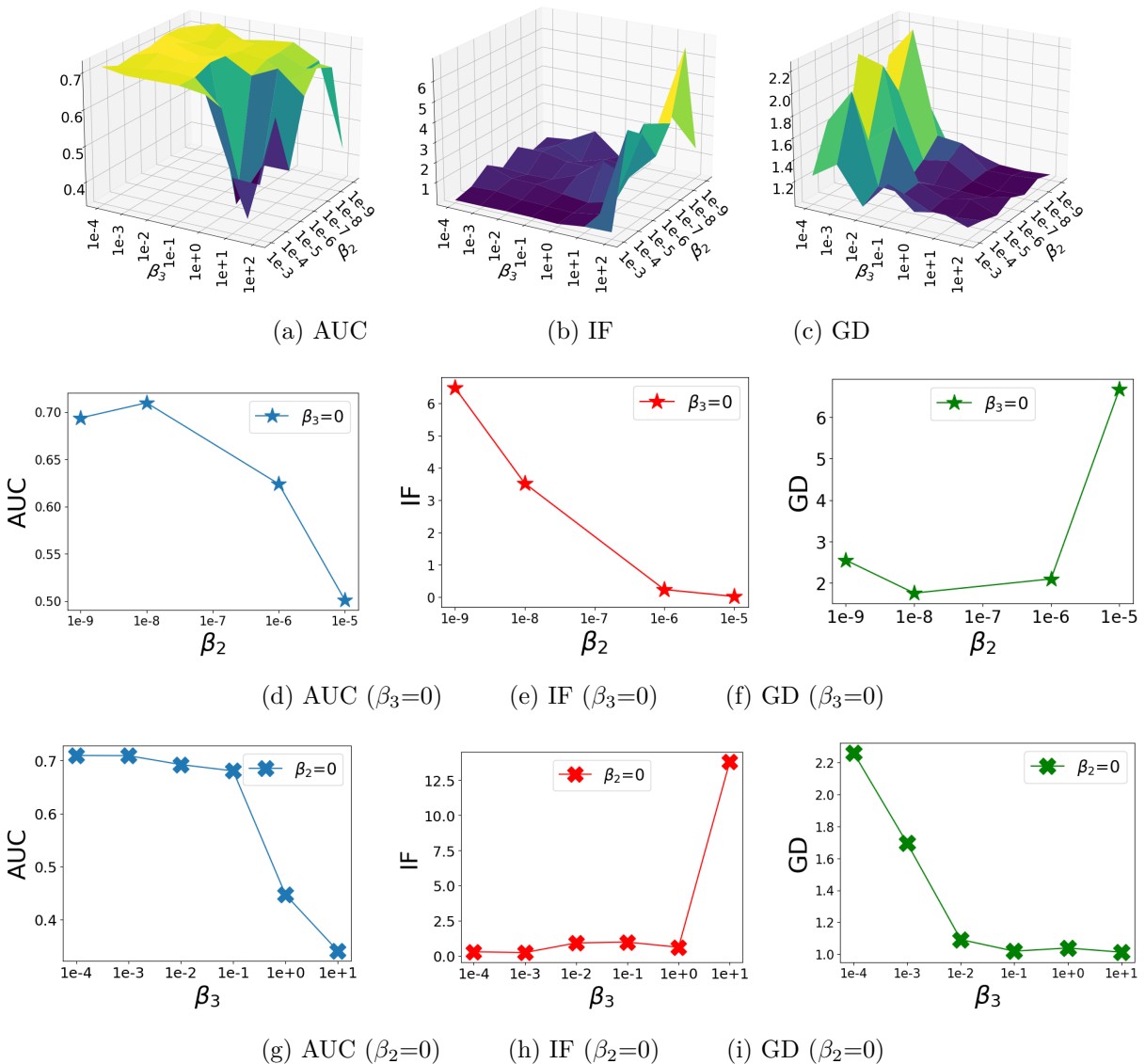

Figure G: Performance results of GRAPHGINI for Credit dataset on GCN-GNN with varying hyperparameters $\beta_2$ for the overall individual unfairness (IF) objective and $\beta_3$ for the group disparity (GD) objective. IF values are in thousands. Figures (a-c) are 3-D representations where the z-axis corresponds to AUC, individual fairness (IF), and group fairness (GD) in figures (a), (b), and (c), respectively. Figures (d-i) show 2-D representations by varying one fairness regularizer and fixing the other fairness regularizer to 0. Here, $\beta_1$ is set to 1 in all figures, and lower IF and GD correspond to better individual and group fairness, respectively.

## O    Sensitivity to Attribute Noise

To evaluate the resilience of GRAPHGINI to perturbations in node features, we perform a controlled attribute-noise sensitivity analysis. To investigate how fairness and performance respond to progressive corruption of input features, while keeping the graph topology fixed. We added Gaussian noise to original feature matrix $X \in \mathbb{R}^{n \times d}$, as follws:

$$\tilde{X} = X + \eta, \quad \eta \sim \mathcal{N}(0, \sigma^2),$$

Table S: Performance of GraphGini compared to Guide with varying homophily levels ($\rho$) on the Income dataset using the GCN backbone. Arrows: ↑ higher is better, ↓ lower is better.

| $\rho$ | Method | AUC (↑) | IF (↓) | GD (↓) | IF-Gini (↓) | GD-Gini (↓) |
|---|---|---|---|---|---|---|
| 0.0 | Guide | 0.72 | 33.3 | 1.12 | 0.26 | 1.12 |
| | GraphGini | 0.72 | 25.4 | 1.10 | 0.18 | 1.10 |
| 0.2 | Guide | 0.73 | 33.1 | 1.12 | 0.26 | 1.11 |
| | GraphGini | 0.73 | 22.3 | 1.05 | 0.17 | 1.09 |
| 0.4 | Guide | 0.74 | 32.2 | 1.08 | 0.26 | 1.11 |
| | GraphGini | 0.74 | 20.8 | 1.03 | 0.16 | 1.06 |
| 0.6 | Guide | 0.74 | 32.7 | 1.02 | 0.25 | 1.10 |
| | GraphGini | 0.74 | 19.7 | 1.02 | 0.15 | 1.05 |
| 0.8 | Guide | 0.75 | 32.7 | 1.03 | 0.20 | 1.10 |
| | GraphGini | 0.75 | 19.4 | 1.00 | 0.15 | 1.05 |

Table T: Performance of GraphGini compared to Guide with varying homophily levels ($\rho$) on the Pokec-n dataset using the GCN backbone. Arrows: ↑ higher is better, ↓ lower is better.

| $\rho$ | Method | AUC (↑) | IF (↓) | GD (↓) | IF-Gini (↓) | GD-Gini (↓) |
|---|---|---|---|---|---|---|
| 0.0 | Guide | 0.72 | 85.2 | 1.20 | 0.27 | 1.19 |
| | GraphGini | 0.72 | 35.1 | 1.18 | 0.23 | 1.15 |
| 0.2 | Guide | 0.73 | 85.2 | 1.20 | 0.26 | 1.19 |
| | GraphGini | 0.74 | 33.5 | 1.13 | 0.22 | 1.14 |
| 0.4 | Guide | 0.74 | 76.4 | 1.19 | 0.25 | 1.19 |
| | GraphGini | 0.74 | 29.9 | 1.09 | 0.21 | 1.13 |
| 0.6 | Guide | 0.74 | 75.2 | 1.19 | 0.25 | 1.19 |
| | GraphGini | 0.74 | 28.2 | 1.06 | 0.21 | 1.12 |
| 0.8 | Guide | 0.75 | 75.2 | 1.19 | 0.25 | 1.19 |
| | GraphGini | 0.75 | 28.1 | 1.04 | 0.20 | 1.11 |

where $\sigma$ controls the noise magnitude. We vary $\sigma \in \{0.0, 0.1, 0.2, 0.3, 0.4\}$ to increase the noise intensity of the features. This approach preserves structural information in the adjacency matrix $A$, ensuring that any observed variation arises solely from feature corruption. It is evident from Tables U, V and W that across all datasets, the performance of GraphGini remains stable for moderate noise levels ($\sigma \leq 0.2$), while the deterioration in fairness metrics under stronger noise.

# P   Sensitivity to Group Imbalance

Handling group imbalance is critical, as real-world graph data often exhibits significant disparities in group representation. In our work, we have carefully selected datasets and corresponding train–test splits to ensure that our method can effectively handle such imbalanced scenarios. Specifically, two of our datasets *Credit* and *Income* are highly imbalanced, whereas the *Pokec-n* dataset is relatively balanced. For clarity, the approximate group ratios in these datasets are summarized in Table X below. Our experimental results (Tables 3 and 4) further demonstrate that GraphGini remains robust and consistent even when the sensitive groups are highly imbalanced, highlighting its stability and fairness-preserving behavior across diverse data distributions.

Table U: Performance of GRAPHGINI compared to GUIDE when noise ($\sigma$) is added to the features on the Credit dataset. Arrows: ↑ higher is better, ↓ lower is better.

| $\sigma$ | Method | AUC (↑) | IF (↓) | GD (↓) | IF-Gini (↓) | GD-Gini (↓) |
|---|---|---|---|---|---|---|
| 0.0 | GUIDE | 0.69 | 1.94 | 1.01 | 0.15 | 1.41 |
| | GRAPHGINI | 0.69 | 0.22 | 1.01 | 0.12 | 1.09 |
| 0.1 | GUIDE | 0.68 | 1.95 | 1.02 | 0.15 | 1.40 |
| | GRAPHGINI | 0.68 | 0.24 | 1.02 | 0.13 | 1.10 |
| 0.2 | GUIDE | 0.67 | 2.01 | 1.04 | 0.17 | 1.40 |
| | GRAPHGINI | 0.67 | 0.26 | 1.04 | 0.14 | 1.11 |
| 0.3 | GUIDE | 0.66 | 2.05 | 1.10 | 0.17 | 1.40 |
| | GRAPHGINI | 0.66 | 0.28 | 1.07 | 0.15 | 1.12 |
| 0.4 | GUIDE | 0.64 | 2.21 | 1.10 | 0.17 | 1.42 |
| | GRAPHGINI | 0.64 | 0.30 | 1.09 | 0.16 | 1.13 |

Table V: Performance of GRAPHGINI compared to GUIDE when noise ($\sigma$) is added to the features on the Income dataset. Arrows: ↑ higher is better, ↓ lower is better.

| $\sigma$ | Method | AUC (↑) | IF (↓) | GD (↓) | IF-Gini (↓) | GD-Gini (↓) |
|---|---|---|---|---|---|---|
| 0.0 | GUIDE | 0.74 | 39.1 | 1.00 | 0.21 | 1.02 |
| | GRAPHGINI | 0.74 | 19.7 | 1.00 | 0.15 | 1.02 |
| 0.1 | GUIDE | 0.73 | 40.2 | 1.02 | 0.23 | 1.05 |
| | GRAPHGINI | 0.73 | 20.8 | 1.02 | 0.16 | 1.04 |
| 0.2 | GUIDE | 0.72 | 45.5 | 1.04 | 0.23 | 1.08 |
| | GRAPHGINI | 0.72 | 22.1 | 1.04 | 0.17 | 1.05 |
| 0.3 | GUIDE | 0.70 | 48.2 | 1.10 | 0.25 | 1.10 |
| | GRAPHGINI | 0.70 | 23.5 | 1.07 | 0.18 | 1.10 |
| 0.4 | GUIDE | 0.68 | 49.4 | 1.10 | 0.27 | 1.12 |
| | GRAPHGINI | 0.68 | 25.0 | 1.09 | 0.19 | 1.11 |

Table W: Performance of GRAPHGINI compared to GUIDE when noise ($\sigma$) is added to the features on the Pokec-n dataset. Arrows: ↑ higher is better, ↓ lower is better.

| $\sigma$ | Method | AUC (↑) | IF (↓) | GD (↓) | IF-Gini (↓) | GD-Gini (↓) |
|---|---|---|---|---|---|---|
| 0.0 | GUIDE | 0.74 | 65.1 | 1.10 | 0.24 | 1.18 |
| | GRAPHGINI | 0.74 | 28.2 | 1.06 | 0.21 | 1.12 |
| 0.1 | GUIDE | 0.73 | 68.2 | 1.10 | 0.26 | 1.19 |
| | GRAPHGINI | 0.73 | 29.6 | 1.07 | 0.21 | 1.13 |
| 0.2 | GUIDE | 0.72 | 74.3 | 1.15 | 0.28 | 1.21 |
| | GRAPHGINI | 0.72 | 31.1 | 1.09 | 0.22 | 1.14 |
| 0.3 | GUIDE | 0.71 | 78.3 | 1.15 | 0.28 | 1.26 |
| | GRAPHGINI | 0.71 | 32.8 | 1.11 | 0.23 | 1.15 |
| 0.4 | GUIDE | 0.69 | 80.2 | 1.16 | 0.29 | 1.26 |
| | GRAPHGINI | 0.69 | 34.9 | 1.13 | 0.24 | 1.16 |

Table X: Group Ratio across datasets. Ratios denote the proportion of the majority to the minority groups.

| Metric | Credit | Income | Pokec-n |
|---|---|---|---|
| Labels | $\{0, 1\}$ | $\{0, 1\}$ | $\{0, 1\}$ |
| Group ratio (majority : minority) | 78 : 22 | 78 : 22 | 51 : 49 |

## Q  Sensitivity to Embedding Dimension

To understand how the representation size affects the fairness–utility trade-off by varying the hidden embedding dimension $h \in \{8, 16, 32\}$ of the GNN encoder used by GRAPHGINI. All other components (losses, similarity $S$, optimization, and data splits) are kept fixed. For all datasets, we performed an ablation study by changing from small to moderate dimensions ($h = 8 \rightarrow 16/32$), and reported the results in Table Y below. We observe that for larger hidden dimensions, the performance of the GNN begins to diminish. All the results in the manuscript are presented with $h=16$ at which fairness–utility is best balanced.

Table Y: Effect of embedding dimension $h$ on the Credit, Income and Pokec-n datasets (GCN backbone). Arrows: ↑ higher is better, ↓ lower is better.

| $h$ | AUC (↑) | IF (↓) | GD (↓) | IF-Gini (↓) | GD-Gini (↓) |
|---|---|---|---|---|---|
| | | | **Credit** | | |
| 8 | 0.67 | 0.27 | 1.02 | 0.14 | 1.10 |
| 16 | 0.69 | 0.22 | 1.01 | 0.12 | 1.09 |
| 32 | 0.66 | 0.29 | 1.08 | 0.15 | 1.11 |
| | | | **Income** | | |
| 8 | 0.72 | 22.8 | 1.03 | 0.17 | 1.10 |
| 16 | 0.74 | 19.7 | 1.02 | 0.15 | 1.08 |
| 32 | 0.71 | 23.2 | 1.08 | 0.18 | 1.10 |
| | | | **Pokec-n** | | |
| 8 | 0.72 | 31.4 | 1.08 | 0.22 | 1.14 |
| 16 | 0.74 | 28.2 | 1.06 | 0.21 | 1.12 |
| 32 | 0.74 | 30.9 | 1.10 | 0.24 | 1.15 |

