# OpenReview forum: "GraphGini: Fostering Individual and Group Fairness in Graph Neural Networks"
_TMLR — Accepted by TMLR_

### Review · Reviewer_Cesd · 2025-08-07

**Summary Of Contributions:**

The paper proposes GraphGini, a methodology to promote fairness in GNN outcomes. In contrast to traditional Lipschitz-based methodologies which overlook the whole distribution of outcomes, the Gini coefficient allows to enforce fairness  throughout the whole range of outcomes.
The authors highlight the different between the two clearly with a toy example.
The Gini coefficient for individual fairness is defined based on the node embeddings, while the group fairness depends on the individual fairness over the nodes in the groups defined by sensitive attributes.
To take into account the Gini coefficient in the loss function, a differentiable upper bound is derived.
The authors use the Nash social welfare to showcase that the fairness among groups is optimised in the GNN outcomes.
Finally the training is viewed as as a multi-objective optimisation using gradient normalisation.
The experiments indicate that GraphGini outperforms 11 fair GNNs on three datasets in terms of fairness without sacrificing too much AUC.

**Audience:**

Yes

**Audience Explanation:**

Researchers working on fairness on graph learning should be interested in this.

**Broader Impact Concerns:**

There is no clear ethical concern about this work.

**Claims And Evidence:**

Yes

**Claims Explanation:**

The authors clarify that the limitation Lipschitz- based methods stem from the fact that it is based on the maximum distance over all node pairs, instead of every node pair, and showcase with a toy example how this might mean that two different models can have the same constant.
The use of Gini coefficient and group fairness is clearly motivated in Observation 1, where starting from the penalty term we end up with the definition of equal opportunity in terms of outcomes.
The upper bound derivation seems correct, although I did not examine App E in all detail.
The Nash social welfare program makes sense in order to extend the individual fairness constraint to group fairness inside the loss function, and the gradient norm is required to optimise overall all regularisations.
Finally the results overall support the advantage of GraphGini compared to the literature in terms of fairness metrics.

**Requested Changes:**

Overall this is a well-written text, with clear motivation, solid methodology and promising results.

My only question pertains to the use of the similarity matrix based on the features.  To my understanding, groups are decided based on the sensitive feature when S is not topological (based on the first sentence in 4.5).
I thus wanted to understand what are the limitations of GraphGini when S is based on the features and not the graph, because, to my understanding, this scenario might be more applicable to the real world. Since the the matrix becomes a dense V^2, I assume there are concerns at least pertaining to space requirements.

Some minor comments:

The code is not modular. In the majority it seems like it is derived by standalone notebooks which makes the code’s reusability and readability hard.

The proof of proposition 3 could be potentially omitted as it is trivial, or at least be put in the appendix.

The proof of proposition 4 is a direct reference to another paper, hence it could could also be omitted.

In the Proof of Proposition 2. App E, there a parentheses missing in “..,w have: min_z”.

In the Proof of Proposition 4 App F, there is a numeric reference while the references is not enumerated.

---

### Review · Reviewer_ZP48 · 2025-08-28

**Summary Of Contributions:**

This paper studies group and individual fairness for graph neural networks (GNNs) while maintaining task utility. It proposes the Gini coefficient as a unified optimization criterion for both individual and group fairness, as an alternative to the Lipschitz constant, and provides an upper bound via the trace of the Similarity Laplacian.

**Strengths**
- The paper is well-organized and clearly written.
- The choice of the Gini coefficient is well motivated, and the theory is rigorous.
- The trace-based upper bound is insightful and helps connect the fairness objective to established spectral tools.

**Weaknesses**
- Missing comparisons to non-Lipschitz approaches, especially trace- or Laplacian-regularization baselines that are closely related to the proposed bound.
- The experimental comparisons in Tables 3–4 are hard to interpret because methods are evaluated at different points on the utility–fairness Pareto front. As presented, it is unclear whether improvements come from a different trade-off choice rather than a strictly better method.

**Audience:**

Yes

**Audience Explanation:**

The theoretical contributions are interesting and shed light on how Gini coefficients relate to the well-known equal opportunity metric in fairness, and the upper bound via the trace of the Similarity Laplacian provides a gradient-based optimization pathway. These points should interest readers working on fairness in GNNs and spectral regularization. Stronger, apples-to-apples empirical validation on standard benchmarks would further broaden the paper’s appeal and practical relevance.

**Broader Impact Concerns:**

This work can improve fairness in ML tasks involving GNNs, which is socially beneficial. At the same time, it could be misused (intentionally or inadvertently) to disadvantage vulnerable demographic groups (e.g., via proxy attributes or poorly chosen utility–fairness trade-offs), so its use should be approached with caution.

**Claims And Evidence:**

Yes

**Claims Explanation:**

The claims are theoretically supported by convincing evidence in Observation 1 and Propositions 1 and 2. However, for real-world fairness applications, the empirical support using GNNs is insufficient and difficult to interpret, e.g., Tables 3–4 compare methods at different points on the utility–fairness Pareto front.

**Requested Changes:**

Please use different values for the $\beta$ coefficients in Eq. (11) to produce multiple points along the utility–fairness Pareto front.

**Details to include:**
- Sweep a sensible range of $\beta$’s (e.g., per-term grids or log-spaced values) for each dataset.
- Plot full utility–fairness curves for GraphGini and baselines; report matched comparisons at equal utility (or equal fairness) to ensure apples-to-apples evaluation.

---

### Review · Reviewer_ycqx · 2025-10-31

**Summary Of Contributions:**

This paper proposes GraphGini, which trains GNNs with fairness regularization by replacing Lipschitz-style individual fairness with a Gini-based dispersion of node embeddings (optimized via a Laplacian surrogate), and adding a group-fairness term balanced through a Nash Social Welfare–like product. The full objective combines task loss, individual fairness, and group fairness, with GradNorm auto-tuning their weights. On several node-classification benchmarks and backbones, it reports improved fairness metrics with minimal accuracy drop.

**Audience:**

Yes

**Audience Explanation:**

Graph fairness is an important problem that many researchers are interested in.

**Claims And Evidence:**

No

**Claims Explanation:**

1. The proposed loss $\mathcal{L}_2$ is unmotivated by Gini and duplicates prior art. This training loss is the same structural surrogate that appears in Eq. 4 of InFoRM [1] for individual fairness on graphs. The proposed view with the Gini index does not bring in any fundamental breakthroughs.
2. The paper argues that the baseline unfairness in Eq. 1 is flawed “because of the max.” But that invites a straightforward design change (e.g., softmax/LogSumExp, p-norm pooling, Huberized max, top-k averaging, or smooth-max) instead of introducing a new metric that, after relaxation, converges to the same Laplacian upper bound as before. Without an ablation showing that alternative pooling fails while the Gini route succeeds, the step to a new metric is unjustified.
3. "Gini is superior to Lipschitz” is only shown via a toy counterexample. The paper’s central conceptual claim rests on a hand-crafted example rather than a theorem (no conditions for equivalence/separation, no bounds that compare the two surrogates, no calibration/consistency result). In the absence of rigorous analysis or stress tests across regimes (dense/sparse graphs, noisy attributes, varying homophily), the superiority claim is not convincing.
4. Furthermore, if we look at the standard definition of the Gini index (Eq. 2), then we can find that the canonical Gini is scalar; the authors generalize this definition in (3) for vector-valued inputs with 1-norm. This comes without sufficient justification. To be specific, any norm in one dimension is essentially of the absolute value form. Then we can directly define it as the 2-norm form, which avoids Proposition 1 and in this regard, past works like [1] are already optimizing the Gini-index objective, completely diminishing the contribution of the paper.
5. The contribution claim of “auto-tuning of multi-objective loss” in 1.2 relies on existing techniques (e.g., GradNorm). It’s an implementation detail, not a new algorithmic contribution; the paper should avoid over-claiming.
6. Final objective (Eq. 8) is a direct, routine combination of losses. The method aggregates standard terms without a new optimization principle or analysis demonstrating an edge over baselines; as presented, it’s too incremental to advance the field.

[1] InFoRM: Individual Fairness on Graph Mining. KDD'20

**Requested Changes:**

1. Justify (or revise) the multivariate Gini-index definition.
2. Provide a formal result showing that the proposed Gini index is better than conventional metrics in some aspects, and why we need a foundational change rather than a pooling-operator change.
3. Differentiate the main methodology from previous arts.
4. Reposition the contribution section to avoid over-claims.
5. Add some sensitivity studies. i.e. vary (a) homophily/heterophily levels (b) attribute noise (c) group imbalance ratio (d) embedding dimension (e) $\beta$ initialization with or without GradNorm and then test the method.

---

> ### Author Response · Authors · 2025-11-14
> **Reply to Reviewer ycqx’s Comments Part-1**
>
> We thank the reviewer for critically evaluating our paper and providing valuable suggestions to improve the manuscript. We have addressed the questions raised by the reviewer and added the required ablations in the updated manuscript accordingly.
>
>
> ### Requested changes
> > **1. Justify (or revise) the multivariate Gini-index definition**
>
> > **2. Provide a formal result showing that the proposed Gini index is better than conventional metrics in some aspects, and why we need a foundational change rather than a pooling-operator change.**
>
>
> We have included the justification, formal proof, and empirical comparison of the proposed method with smooth surrogates in Appendix K of the revised manuscript (along with a pointer after Definition~6). GraphGini consistently outperforms both the softmax and top-$k$ surrogates, indicating that modifying pooling surrogates alone is insufficient to achieve fair graph embeddings.
>
> > **3. Differentiate the main methodology from previous arts.**
>
> **Response:** There are three concrete differences, as also highlighted in our contributions section.
>
> * **Joint optimization of both individual and group fairness:** Most works (such as Inform) target either only individual fairness or group fairness. This is the only work other than the Guide (Song et al. 2022) to target both.
>
> * **Theoretical grounding:** Compared to Guide, our loss function is different. To capture fairness on the entire spectrum of outcomes, we propose the Gini coefficient, a well-established social welfare metric (Propositions 1 and 2).
>     - Our formulation reinterprets individual fairness as **minimizing distributional disparity** across node embeddings, rather than merely bounding the worst-case deviation. This yields a convex and differentiable objective that unifies individual and group fairness within a single Gini-based framework, making it theoretically grounded rather than heuristic. **We observe its empirical impact on improving fairness metrics across all our experiments.**
>     - We show that the Gini coefficient offers a more robust and holistic fairness measure, naturally leading to *equal opportunity*. Guide does not provide guarantees of equal opportunity. Our response to Q1 and 2 above, further elaborates on how Gini is superior in characterizing the entire spectrum of outcomes when compared to Lipschitz constant or alternative pooling operators. **This is also supported empirically in Table 8.**
>
> * **Pareto-optimality:** The inclusion of the Nash Social Welfare component ensures Pareto-optimality among groups (Proposition 3).
>
> > **4. Reposition the contribution section to avoid over-claims.**
>
> **Response:** We have removed the bullet point on leveraging GradNorm as a contribution, as suggested (section 1.2). While the current version already cites GradNorm appropriately and limits our claim to its application for balancing individual and group fairness, we acknowledge the feedback and will remove this from the explicit list of contributions to avoid any overstatement.

---

> > ### Author Response · Authors · 2025-11-14
> > **Reply to Reviewer ycqx’s Comments Part-2**
> >
> > ### 5. Ablation Study
> >
> > **Response:** We have included detailed ablation studies on the sensitivity of **GraphGini** to (i) homophily/heterophily, (ii) attribute noise, and (iii) embedding dimension in Appendices~N, O, and Q, respectively, in the revised manuscript.
> >
> > For homophily sensitivity, we follow a degree-preserving edge-rewiring strategy. For each homophily level $\rho \in {0, 0.2, 0.4, 0.6, 0.8}$, a fraction $\rho|E|$ of edges is randomly selected, and one endpoint of each edge is reassigned to a node sharing the same class label as the source node. For attribute-noise sensitivity, we study the robustness of GraphGini under controlled perturbations of node features, gradually increasing the corruption level while keeping the graph topology fixed. This allows us to isolate the effect of noisy attributes on fairness and predictive performance. For embedding-dimension sensitivity, we examine how the representation size impacts the fairness–utility trade-off by varying the hidden dimension $h \in {8, 16, 32}$ of the GNN, while keeping all other components (loss functions, similarity $S$, optimization settings, and data splits) unchanged.
> >
> > #### Sensitivity to Group Imbalance
> > We thank the reviewer for explicitly raising this important point. Handling group imbalance is indeed critical, as real-world graph data often exhibits significant disparities in group representation. In our work, we have carefully selected datasets and corresponding train–test splits to ensure that our method can effectively handle such imbalanced scenarios. Specifically, two of our datasets *Credit* and *Income* are highly imbalanced, whereas the *Pokec-n* dataset is relatively balanced. Our experimental results (Tables~3 and~4 in the paper) further demonstrate that **GraphGini** remains robust and consistent even when the sensitive groups are highly imbalanced, highlighting its stability and fairness-preserving behavior across diverse data distributions. These additional details have been incorporated into Appendix~P of the revised manuscript.
> >
> >
> > ####  $\beta$ Initialization with or without GradNorm
> > We would like to clarify to the reviewer that this ablation is already included in the main manuscript (see Table~5). Specifically, the results compare models trained with fixed $\beta$ initialization against those using GradNorm-based adaptive balancing. The table demonstrates that GradNorm provides more stable and consistent optimization across datasets, validating our choice to include it in the final framework.
> >
> > We sincerely thank the reviewer for their valuable suggestions. We believe that we have incorporated all the requested changes and that these additions have significantly improved the clarity and overall quality of the manuscript.

---

> > > ### Author Response · Authors · 2025-11-14
> > > **Reply to Reviewer ycqx’s Comments Part-3**
> > >
> > > ## Answer Explanations
> > > #### [1-4]
> > > We again thank the reviewer for the careful reading and for highlighting several important aspects of our work. Comments~1-4 collectively pertain to our formulation of the Gini coefficient as an individual fairness measure. We address these concerns below in a unified and comprehensive manner.
> > >
> > > We respectfully disagree with the reviewer’s comment. Although our term $L_{2} = -\mathrm{Tr}(Z^{\top}LZ)$ may appear structurally similar to the regularizer used in InFoRM, our formulation is motivated from a fundamentally different perspective.
> > >
> > > We derive $L_{2}$ from the well-established notion of the Gini coefficient, a principled measure of inequality in the economics literature, and extend it to the graph learning setting to quantify representational disparity among node embeddings.
> > >
> > > In contrast, InFoRM heuristically interprets the same trace form as an approximation of the **Lipschitz constant**, which captures only the **worst-case** disparity and is generally unknown or intractable to compute for GNNs. As the authors of InFoRM note,
> > >
> > > “Eq.~(1) can be interpreted from the perspective of the Lipschitz constant,” highlighting this fact.
> > >
> > > In GraphGini, we derive $L_{2}$ as a **differentiable convex upper bound** of the Gini coefficient, a distributional fairness measure that accounts for disparities across all node pairs. Moreover, InFoRM addresses only individual fairness, whereas GraphGini simultaneously incorporates group fairness. Also, replacing the $\max$ operator in Eq. (1) with smooth variants (e.g., softmax, $p$-norm, or Huberized max) would still operate on the **worst-case pairwise disparity** and remain limited to local sensitivity, thereby inheriting the same limitations of the Lipschitz-based formulation. In contrast, our use of the Gini coefficient introduces a **distribution-level fairness measure** that captures inequality across **all** node pairs rather than only extremal ones. This shift is not a numerical smoothing trick but a conceptual re-formulation that connects GNN fairness to well-established economic inequality theory, yielding provable robustness (Propositions 1–2) and empirical gains (Tables~3–4). Hence, the Gini route is theoretically motivated and provides broader expressivity beyond any soft-approximation of the max operator.
> > >
> > >
> > > The illustrative example was intended to offer intuition, not to serve as the sole justification of Gini’s superiority, but rather the result provided in Propositions 1–2, formally establishes the differentiable convex upper bound of the Gini coefficient and proves equivalence in minimisers with the true non-differentiable form. Furthermore, our extensive empirical evaluations (Tables~ 3, 4, 8). Below, we also tested our method across different regimes by varying homophily and attribute noise, and its performance is consistent. We will add to the ablation study of the manuscript.
> > >
> > > The extension from the scalar Gini index (Eq. 2) to the vector-valued form (Eq. 3) is a necessary generalization for GNNs, where each node embedding $z_i \in \mathbb{R}^c$ is multidimensional. The $\ell_1$ -based formulation preserves the additive interpretation of disparities across embedding dimensions, remaining faithful to the classical notion of inequality. Proposition 1 further shows that the $\ell_2$-based trace term is a convex upper bound of this multidimensional Gini formulation—a connection absent in prior works such as InFoRM. Thus, our formulation extends, rather than replicates, the classical Gini objective for graph embeddings.
> > > #### [5]
> > > We fully agree that GradNorm is an existing technique and do not claim it as a novel algorithmic contribution. The contribution lies in its integration within our fairness framework, where utility, individual fairness, and group fairness operate on different scales and tend to destabilize training. Prior work (e.g., GUIDE) requires manual hyperparameter tuning, whereas incorporating GradNorm enables automatic, stable, and scale-invariant balancing. This is a system-level contribution, not an optimization novelty.
> > > #### [6]
> > > While Eq. (11) combines three losses, the novelty is their principled unification. GraphGini introduces:
> > > (i) a Gini-based convex surrogate with provable equivalence to the original non-smooth formulation,
> > > (ii) Nash Social Welfare–based coupling ensuring Pareto fairness across groups, and
> > > (iii) automatic gradient balancing for reliable multi-objective convergence.
> > > Together, these yield a coherent and theoretically grounded fairness framework, not a routine weighted sum of losses.

---

### Author Response · Authors · 2025-11-14
**Updated version of the manuscript after addressing all reviewer comments and suggestions.**

We thank the reviewers for their comments. Based on the comments, we have revised the paper and submitted a revised version. In the rebuttal below, we will detail how the revisions address the issues raised by the reviewers. For ease of reading, we have coloured the newly added text in blue in the revised version.


Regards,

The Authors

---

### Decision · Action_Editor_A3wE · 2025-12-19

**Recommendation:** Accept with minor revision

**Additional Comments:**

As the minor revision, the authors should address the five issues raised above regarding evidence.  Additionally, the paper the paper should be finalized by removing the blue text indicating changes and doing an additional copy-editing pass with particular emphasis on the changes during the review process since I spotted a couple of issues that were introduced:
- 1.2 still refers to four key innovations where there are now three
- 3.5 now starts with "Convergencey" which I assume is meant to be Convergence.

As a process note, some of the authors' responses appear to have accidentally not included the reviewers as readers, so the reviewers will not have seen them, which lead one reviewer to believe the questions had not been addressed.  However I have reviewed the concerns and the response and am satisfied.

**Audience:**

Yes

**Audience Explanation:**

All three reviewers agree that the topic is of interest to those interested in fair GNNs.  Some comment on their appreciation of the intuitions and connections provided by the theoretical results.  The empirical results demonstrate strong performance suggesting interest for those seeking practical engineering insights.

**Claims And Evidence:**

Yes

**Claims Explanation:**

While there were a number of issues raised in the initial reviews, through substantial additions and revisions the authors have addressed many of these.  After these, two of the three reviewers are satisfied by the quality of evidence presented.  One raises what I believe are valid concerns about the evidence on certain points.  However, as I detail below I believe these are all addressable with minor modifications to enable the paper to meet TMLR's standard.

One issue is around the novelty of the term Tr(Z^\top L Z), which I agree with the reviewer does appear to have been used in prior work such as InFoRM.  Where I differ with the reviewer is that I believe it is a fine contribution to reuse it as part of a larger system with new theoretical motivation and connections.  However, if this is the case the paper needs to be more explicit that the analysis and not the loss itself is the new contribution with appropriate citations to prior work.  There is a bit of ambiguity about this in the discussion where the authors commented that this loss "may appear structurally similar" implying that it is actually different in some way.  If I am incorrect about it being the same loss then an explanation is needed as to what the difference is.

A related issue is the relative merit of Proposition 1 and the current analysis relative to the reviewer's proposal of directly defining the vector-valued version using the 2-norm.  While I am not sure I agree, I credit the reasonableness of the authors' position that the 1-norm version better captures the desired notion of fairness.  But I believe this logic and the alternative should be included in the paper (e.g. as a footnote) since some readers may find this version of the connection more convincing.

Relatedly, I'm not entirely convinced that Proposition 1 is correct as stated.  The proof includes the claim that "the denominator in Eq. 3 only scales the numerator and does not affect the inequality."  But can't scaling the numerator by a sufficiently small constant (less than 1) cause the inequality to be violated?  If there is some reason this doesn't happen the explanation should be made clearer.  Otherwise the claim could simply be restated to explicitly include the denominator since I agree this is largely irrelevant to the point of the proposition.

Proposition 2 currently lacks any motivation or discussion, which should be added to justify its inclusion.  I am not sure what this would look like for the third claim of Proposition 2 which seems true but irrelevant to me.  Since this is one of three component losses it will never be exactly minimized, so I don't see why a claim about what would happen if we did is relevant.  Indeed, as the proof shows the minimizers are trivial.  So unless this can motivated for the reader this part of the proposition should be removed.\\

Assuming these four issues are addressed, all of which require only minor revisions, I am satisfied with the evidence for this portion of the paper.  The reviewer raises one additional concern about extent to which the combination of losses really centers Gini, but since both fairness losses are motivated in terms of Gini, I am satisfied with the evidence in that regard.

One final separate issue is that the claims in the conclusion are a bit strong and should be toned down.  In particular there is no real evidence that "This particular way of using Gini is likely to have a major impact on future research".  A version claiming it "may be of interest" would be more measured.  Given that the use of GradNorm was removed from the list of major contributions, the sentence "Unlike existing state-of-the-art methods, the GraphGini automatically balances all three optimization objectives—utility, individual fairness, and group
fairness—eliminating the need for manual tuning of weight parameters." should either be removed as well or rephrased to make it about the the effectiveness of GradNorm as a technique in this domain.